# Natural selection and genetic diversity maintenance in a parasitic wasp during continuous biological control application

Bingyan Li [1,4], Yuange Duan [1,4], Zhenyong Du[1], Xuan Wang[1], Shanlin Liu [1], Zengbei Feng[1], Li Tian[1], Fan Song[1], Hailin Yang[2], Wanzhi Cai[1], Zhonglong Lin [3] ✉ & Hu Li [1] ✉

*Aphidius gifuensis* is a parasitoid wasp and primary endoparasitoid enemy of the peach potato aphid, *Myzus persicae*. Artificially reared, captive wasps of this species have been extensively and effectively used to control populations of aphids and limit crop loss. However, the consequences of large-scale releasing of captive *A. gifuensis*, such as genetic erosion and reduced fitness in wild populations of this species, remains unclear. Here, we sequence the genomes of 542 *A. gifuensis* individuals collected across China, including 265 wild and 277 human-intervened samples. Population genetic analyses on wild individuals recovered Yunnan populations as the ancestral group with the most complex genetic structure. We also find genetic signature of environmental adaptation during the dispersal of wild populations from Yunnan to other regions. While comparative genomic analyses of captive wasps revealed a decrease in genetic diversity during long-term rearing, population genomic analyses revealed signatures of natural selection by several biotic (host plants) or abiotic (climate) factors, which support maintenance of the gene pool of wild populations in spite of the introduction of captive wasps. Therefore, the impact of large-scale release is reduced. Our study suggests that *A. gifuensis* is a good system for exploring the genetic and evolutionary effects of mass rearing and release on species commonly used as biocontrol agents.

Biological control (or biocontrol in short) is the utilization of natural enemies or biocontrol organisms (such as arthropods, microorganisms, and invertebrates) to control populations of agricultural pests[1,2]. Among biocontrol approaches, augmentative biocontrol is an ideal and effective long-term strategy instead of pesticides to ensure successful pest control, using mass culture and the periodic release of biocontrol agents are required to ensure successful pest control[3–5], although there are examples of successful biological control programs of aphids based on single introduction events with no further releases of biocontrol agents[6].

For augmentative biocontrol strategies, individuals are generally captured from native or alien populations and reared indoors to meet the demands for immediate or longer-term control of a pest[7]. Parasitoid wasps, as highly effective biocontrol agents, have been extensively applicated, and some can be easily reared under industrial conditions[8–10]. The mass rearing of parasitoid wasps could lead to the loss of host fidelity, poor parasitoid performances and population degradation after generations of breeding[11,12]. Besides, the introduced natural enemies for augmentative biocontrol would undergo inbreeding depression and bottlenecks, consequently reduced genetic

[1]Department of Entomology and MOA Key Lab of Pest Monitoring and Green Management, College of Plant Protection, China Agricultural University, Beijing 100193, China. [2]Tobacco Company, Yuxi 653100, China. [3]Yunnan Tobacco Company of China National Tobacco Corporation, Kunming 650011, China. [4]These authors contributed equally: Bingyan Li, Yuange Duan. ✉e-mail: sdlzl1983@163.com; tigerleecau@hotmail.com

variants of the bred population[13–15]. However, individuals from long-term rearing systems typically exhibit low fitness or degenerate traits compared to their wild counterparts[16] as a result of genetic erosion caused by founder effects, inbreeding, or gene drift resulting from a small effective population size[17].

Long-term extensive release of artificially reared populations may potentially pose negative effects on wild populations[18]. The gene pool of wild populations can be infiltrated by that of the released populations, resulting in outbreeding depression[18], which has been found in some plants or animals like *Eucalyptus nitens*[19], *Salmo trutta*[20]. On the other hand, natural selection may serve as a safeguard to preserve the genetic diversity of wild populations[21]. Balancing selection is a known mechanism that maintains genetic polymorphism in populations for longer even if the populations is small[22]. Long-term balancing selection shaped the genetics of immune systems in the genus *Capsella* that ancestral variation preferentially persists at immunity related loci, and as the main diver of genomic variability after a population bottleneck[23]. Comparing mass culture stocks of *Drosophila ananassae* with their natural populations, it was observed that there were fewer heterozygotes in mass culture stocks, and none of the populations were found to become monomorphic for all allozyme loci suggesting the balancing selection at allozyme loci[24]. However, studies on the effects of biocontrol agent releases on their corresponding wild populations are surprisingly sparse in the current global context of increased demands for Integrated Pest Management. This poses a glaring knowledge gap that urgently needs to be addressed. Moreover, upon the release of mass-reared populations, the extent to which natural selection can maintain the gene pool of a wild population remains unknown.

*Aphidius gifuensis* Ashmead (Hymenoptera: Braconidae) is an endoparasitoid wasp specializing in aphids[25], and distributes widely in Asian, such as China, Japan and Korea[26–28]. It has been used as a biological control agent on tobacco and vegetables in China and Japan, which was sourced from indigenous wild populations, subjected to large-scale rearing until sufficient individuals were attained for release into open fields[29,30]. In China, this parasitoid wasp has been utilized to control green peach aphid populations in Yunnan Province since 1997[26,31]. For more than two decades, it has been commercialized and successfully reared indoors on the green peach aphids, *Myzus persicae* as the host, using tobacco (*Nicotiana tabacum*) as the host plant. The captive individuals are extensively released into open fields during seasons with high aphid infestations each year in 24 provinces of China[32–35]. Long-term mass rearing *A. gifuensis* populations has performed degraded phenomenon, such as rapid population degradation, decreased fecundity, and reduced parasitoid performance[36]. Increasing and continuous release of captive *A. gifuensis* performing low fitness and degenerate traits poses a potential risk to the genetic diversity and variants of wild populations and ecosystems. There is an urgent need to understand how the long-term field application of a biological control agent could affect the genetic diversity of wild populations.

In this study, we generate a high-coverage whole-genome resequencing dataset of 542 *A. gifuensis* individuals, including wild populations across their natural range, captive-bred populations and populations from experimental tobacco fields after release (post-release populations). We aim to investigate the following specific questions: (1) What is the contemporary status of *A. gifuensis* wild populations in terms of population structure and genetic diversity? (2) Could we find genetic basis for local adaptation? (3) What extent of genetic erosion is potentially associated with the long-term mass rearing? and (4) Could natural selection maintain the gene pool of wild populations upon the release of artificially bred populations? Through population genomic approaches, we answer the above questions. Firstly, we discover that the Yunnan populations are the ancestral group and have the most complex genetic structure. We then find

genetic evidence for environmental adaptation during the dispersal of wild populations from Yunnan Province to the entire country. Comparative genomic analyses reveal a decrease in genetic diversity during long-term mass rearing. However, compared to the putatively neutral synonymous sites, those functionally important sites (such as nonsynonymous sites) in wild populations are likely to be preserved by natural selection. We demonstrate that natural selection maintains the gene pool of wild *A. gifuensis* populations, thereby reducing the impact of large-scale releases. In conclusion, our study reveals the efficacy and safety of using *A. gifuensis* as a biocontrol agent.

## Results

### Population genomic resequencing datasets

The genomes of 265 wild *A. gifuensis* female adults, widely distributed across China, particularly in Yunnan Province, were sequenced (Supplementary Fig. 1 and Supplementary Data 1). In addition, we sequenced human-intervened populations, including 160 individuals from captive-bred populations and 117 individuals from local experimental tobacco fields with long-term and ongoing application of reared *A. gifuensis* (Supplementary Fig. 1 and Supplementary Data 2). Accordingly, the three groups were described as wild (N), artificially bred (B), and post-release (T) populations. Through Illumina NovaSeq sequencing, we generated ~4.20 Tb paired-end clean data in total, yielding an average depth of 40.63× and average genome coverage of 97.36% per individual after genome mapping onto an *A. gifuensis* reference (GenBank: GCA_014905175.1) (Supplementary Data 3). Twenty-eight individuals from the wild populations were identified as full- or half-siblings of other individuals (Supplementary Data 4) and were excluded from subsequent analysis. After stringent filtering, 1,865,574 SNPs were retained for downstream analyses.

We additionally assembled whole mitochondrial genomes (mitogenomes) for all individuals from the resequencing data to investigate their maternal inherent patterns, resulting in 542 full mitogenomes (37 genes) with assembly lengths ranging from 14,647 to 14,671 bp.

### Wild populations in Yunnan display highest genetic diversity

The Bayesian Analysis of Population Structure (BAPS) analysis of mitogenomes identified six distinct genetic lineages in the wild populations (Fig. 1a). Individuals from all clusters except Cluster 4 occurred in Yunnan Province, and those from Clusters 1 and 2 were found only in Yunnan Province. Those from Clusters 5 and 6 were widely spread throughout most of their natural range, from southwest to northeast China. Phylogenetic analysis, inferred using another *Aphidius* species as an outgroup, revealed that Clusters 1 and 2 formed a basal lineage with respect to other genetic lineages, indicating that the Yunnan population (YUN) possessed more ancient traits than other populations (Fig. 1b). In addition, the diversification patterns of the mitogenome haplotypes, in which the YUN populations were more diverse than the others, indicated that *A. gifuensis* could have originated from southwest China and spread its distribution range to the northern and eastern regions.

ADMIXTURE and principal component analysis (PCA) based on genome-wild nucleic SNPs revealed weak genetic differentiation among the non-Yunnan populations, all of which can be treated as a single, undifferentiated population group. Samples from Yunnan Province displayed higher isolation levels, forming three distinct genetic groups: (i) YUNN06 (BSLYN), (ii) YUNN07_1 (YNPEN), and (iii) YUNN01 (CXNHN), YUNN04 (HHMLN), YUNN07_2 (PEMJN), YUNN05 (QJQLN), YUNN02 (WSWTN), and YUNN03 (YNZTN) (Fig. 1c, d and Supplementary Fig. 2). The maximum-likelihood (ML) phylogenetic tree indicated that the YUN populations were divided into two subgroups and represented an earlier split compared to the peripheral populations (Supplementary Fig. 3). The pairwise fixation index ($F_{ST}$), based on whole-genome SNPs between populations sampled at

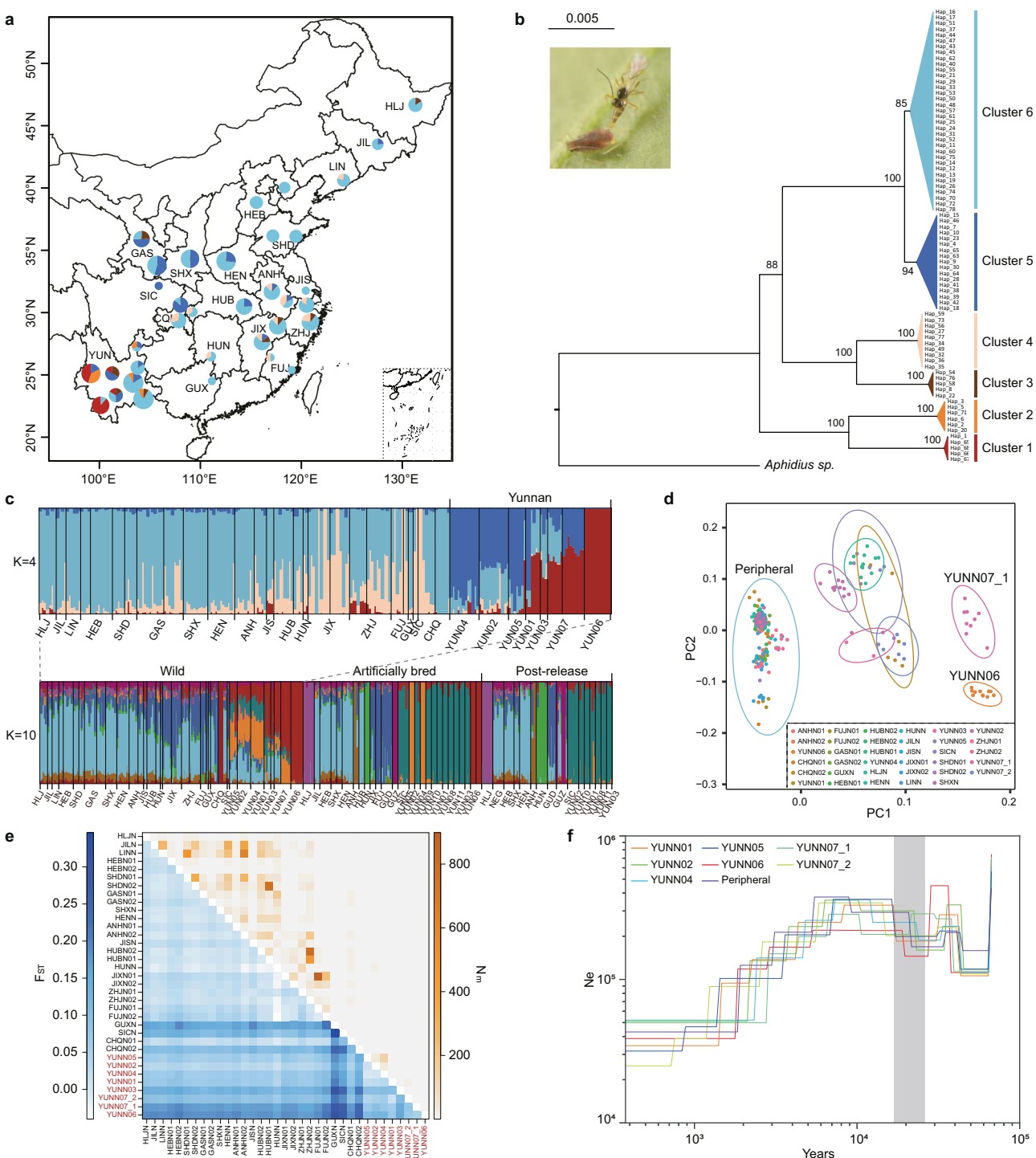

**Fig. 1 | Population structure and demographic history of wild *A. gifuensis* populations. a**, **b** were based on mitogenomes and (**c**–**f**) were based on nucleic SNPs. **a** Clustering of individuals estimated by BAPS. The areas of different colors indicate the proportion of each cluster in the population. The map was drawn by the R Packages maps (https://cran.r-project.org/web/packages/maps). **b** Maximum-likelihood (ML) phylogenetic tree with 1000 bootstraps. The color of each cluster on the phylogenetic tree is consistent with BAPS. **c** Population genetic structure of wild populations (upper) and all populations (bottom) inferred by K-values with the lowest cross-validation error rate of ADMIXTURE. **d** PCA analysis of wild populations. **e** Population genetic differentiation (lower triangle) and corresponding gene flow Nm (upper triangle). The populations of Yunnan are marked in red. **f** Demographic history of wild populations predicted by SMC + +. The gray shadow denotes the last glacial period, approximately 11.5−20 K years ago. Source data are provided in the Source Data file. The detail information of each population abbreviations is listed in Supplementary Data 1 and Supplementary Data 2.

different locations, ranged between −0.0400 and 0.3462 (Supplementary Data 5). The $F_{ST}$ values between the YUN populations (especially YUNN06 and YUNN07_1) and peripheral populations were significantly higher than those between peripheral populations (Fig. 1e).

SMC + + revealed similar pattern of population effective size ($N_e$) changes of major geographical population groups, with two population expansions and two population bottlenecks (Fig. 1f and Supplementary Fig. 4). The population YUNN06 and YUNN07_1 expanded, whereas other YUN populations and the peripheral subgroup

experienced a temporary decrease during the Last Glacial Maximum (Fig. 1f). Thereafter, populations expanded and attained their peak at the end of the last glacial period (11.5–20 Ka), after which they began to decline around 2000–7000 years ago.

## Genetic changes associated with geographic dispersal across changing environment

The environment of different geographic *A. gifuensis* populations varies dramatically. For instance, population Heilongjiang (HLJN) and Yunnan (YUNN) inhabit in different altitude with distinct climate factors (Supplementary Data 6). These indicate that the dispersal of this insect maybe associated with adaptations to distinct local environments. We calculated the allele frequency (AF) for each SNP site within each wild population, to investigate the genetic variations among different geographic populations and any changes associated with the environmental adaptation of *A. gifuensis* during its dispersal. First, we calculated Spearman's correlation (*Rho*) between the AFs and the corresponding longitude or latitude of the sampling regions. The climate factors are almost location-dependent. Thus, we initially examined the potential correlation between mutation spectrum and geographic location, and then expanded the analysis to consider other relevant factors. At $FDR < 0.05$, 276,122 and 285,527 SNPs were significantly correlated with longitude and latitude, respectively (Fig. 2a). Generally, the absolute value of *Rho* (for these significant SNPs) was greater than 0.5, suggesting a strong correlation (Fig. 2a). A gene evm.model.Contig2.505, which functions in the oxidation-reduction process, transport, and catabolism, displayed five non-synonymous SNPs that were significantly correlated with longitude and latitude (Fig. 2b), indicating that it might be relevant to the geographical adaptation of wild *A. gifuensis*. To show how unexpected it is to obtain five (significant) nonsynonymous SNPs in a gene, we first displayed the distribution of CDS (coding DNA sequence) lengths of all genes with significant nonsynonymous SNP (Fig. 2c). The CDS length of gene evm.model.Contig2.505 was almost at the median level (Fig. 2c, 1431 bp, 43.2% quantile). Then we chose the genes with similar CDS length (1000–2000) and calculated how many nonsynonymous SNPs a gene had. In most cases, one gene only had one such SNP (Fig. 2c). This suggests that five such SNPs in evm.model.Contig2.505 should be an extreme value compared with the size of the gene.

Then, to determine whether the SNPs highly correlated with longitude or latitude were more likely to be functional, we ranked the SNPs with $FDR < 0.05$ according to their |*Rho*| values. A higher |*Rho*| value indicated a stronger correlation. SNPs with higher |*Rho*| values exhibited higher nonsynonymous/synonymous (Nonsyn/Syn) ratios (also known as $P_N/P_S$) (Fig. 2d). Since synonymous sites are generally neutral while nonsynonymous mutations are functional, the SNPs participating in environmental adaptation (those with high |*Rho*| values) of *A. gifuensis* were indeed more functional than the uncorrelated SNPs. Besides, genetic load during spatial expansion might also lead to the increase of Nonsyn mutations. To assess whether adaptation or genetic load shaped the Nonsyn/Syn ratios, we calculated the LD ($r^2$) between a Nonsyn SNP and its nearest Syn SNP in pooled wild populations (Fig. 2e). Null hypothesis was, if some observed Nonsyn mutations were resulted from the accumulation of genetic load rather than adaptation, then the LD ($r^2$) between the Nonsyn site and another neutral (Syn) sites should be random and low (Fig. 2f), and should not show differences across different groups of mutations. Conversely, if part of the Nonsyn SNPs were positively selected during dispersal, the LD around the selected Nonsyn sites should be stronger than other non-selected Nonsyn sites (Fig. 2f). Indeed, when dividing the Nonsyn SNPs into those categories as defined in Fig. 2c, the LD ($r^2$) was actually increasing with |*Rho*| values (Fig. 2e), which was rejected the null hypothesis of random accumulation of genetic load. Instead, a more plausible explanation was, the excessive Nonsyn mutations in the "|*Rho*| >0.7" groups were accumulated due to positive selection. Upon

rapid positive selection (e.g., environmental adaptation), the neutral sites (Syn SNPs) around the positively selected Nonsyn SNPs might undergo hitchhiking and show strong linkage with the Nonsyn SNPs. For the less functional or unselected Nonsyn SNPs, their surrounding neutral sites might not show strong linkage with them. Importantly, it could be argued that bottleneck may also result in strong LD pattern, which can appear as genomic blocks or even be strong across the whole genome (Fig. 2f). But this is completely different from our observation that the regions of strong LD were scattered and appeared to be dependent on the correlation with geographical parameters (Fig. 2d, e). Taken together, these results further support our assumption that the Nonsyn SNPs significantly correlated with longitude/latitude (with high |*Rho*| ) are likely to be functional and participating in the environmental adaptation during population dispersal.

We further investigated the associations between genomic variations and climatic variables in different regions. In addition to longitude and latitude, we sought significant correlations ($FDR < 0.05$) between SNPs (AF) and each bioclimatic variable in the WorldClim2 database (Supplementary Data 6). Bio3 (isothermality), Bio4 (temperature seasonality S.D. × 100), Bio7 (annual temperature range), and Bio11 (mean temperature of the coldest quarter) had the greatest number of associated SNPs (Fig. 3a). These bioclimatic variables were highly correlated (Fig. 3a), indicating that these local temperature features might collectively act as selective pressures shaping the SNP landscape. Then, we profiled the correlation coefficient |*Rho*| along the genome, and found that |*Rho*| values between AF and various bioclimatic variables showed similar profiles (Fig. 3b). In particular, an 8 Mb region on chromosome LG06 exhibited extraordinarily high |*Rho*| values for all features (Fig. 3b). Although many of the associated SNPs were located in non-coding or regulatory regions, this pattern suggested that these regions might contain multiple essential genes related to adaptation to temperature. Gene ontology (GO) analysis revealed that the genes within the 8 Mb region of LG06 were enriched in pathways associated with stress response, immunology, neurons, and signaling receptors (Fig. 3c). These functional categories might be essential for organisms to adapt to new environments with completely different climates.

## Genetic erosion emerged during long-term artificial rearing

We compared the population structure and genetic diversity between human-intervened populations (including artificially bred and post-release populations) and wild populations to assess the effect of mass rearing on genetic diversity. ADMIXTURE and PCA results of artificially bred and post-release populations also revealed that the YUN populations were distinct from the peripheral populations (Fig. 1c and Supplementary Fig. 5). The different structure pattern between YUN and the peripheral population was similar to wild population. Artificially bred populations that followed the same introduced breeding origin had similar genetic structure, such as SICB and YUNB09 (YNYXB) populations. In addition, the results indicated that different genetic structure existed only in human-intervened populations but not in wild populations, such as populations HLJB, ANHB, HUNB, and GUZB, which exhibited differentiation from the others.

The genetic diversity (π) of artificially bred populations was significantly lower than that of the wild populations ($\pi_{artificially\ bred} < \pi_{wild}$, $P < 0.05$), and the post-release group had a genetic diversity level between the wild and the artificially bred groups (Fig. 4a and Supplementary Data 7). Furthermore, according to Weir and Cockerham's $F_{ST}$ estimator, the maximum pairwise-$F_{ST}$ was observed between the artificially bred and wild groups (mean pairwise-$F_{ST} = 0.0121$), which was slightly higher than the pairwise-$F_{ST}$ between the post-release and wild groups (mean pairwise-$F_{ST} = 0.0089$). Genetic differentiation between the artificially bred and post-release groups was the smallest (mean pairwise-$F_{ST} = 0.0021$). For each sample location, we observed a similar trend among different populations in most regions

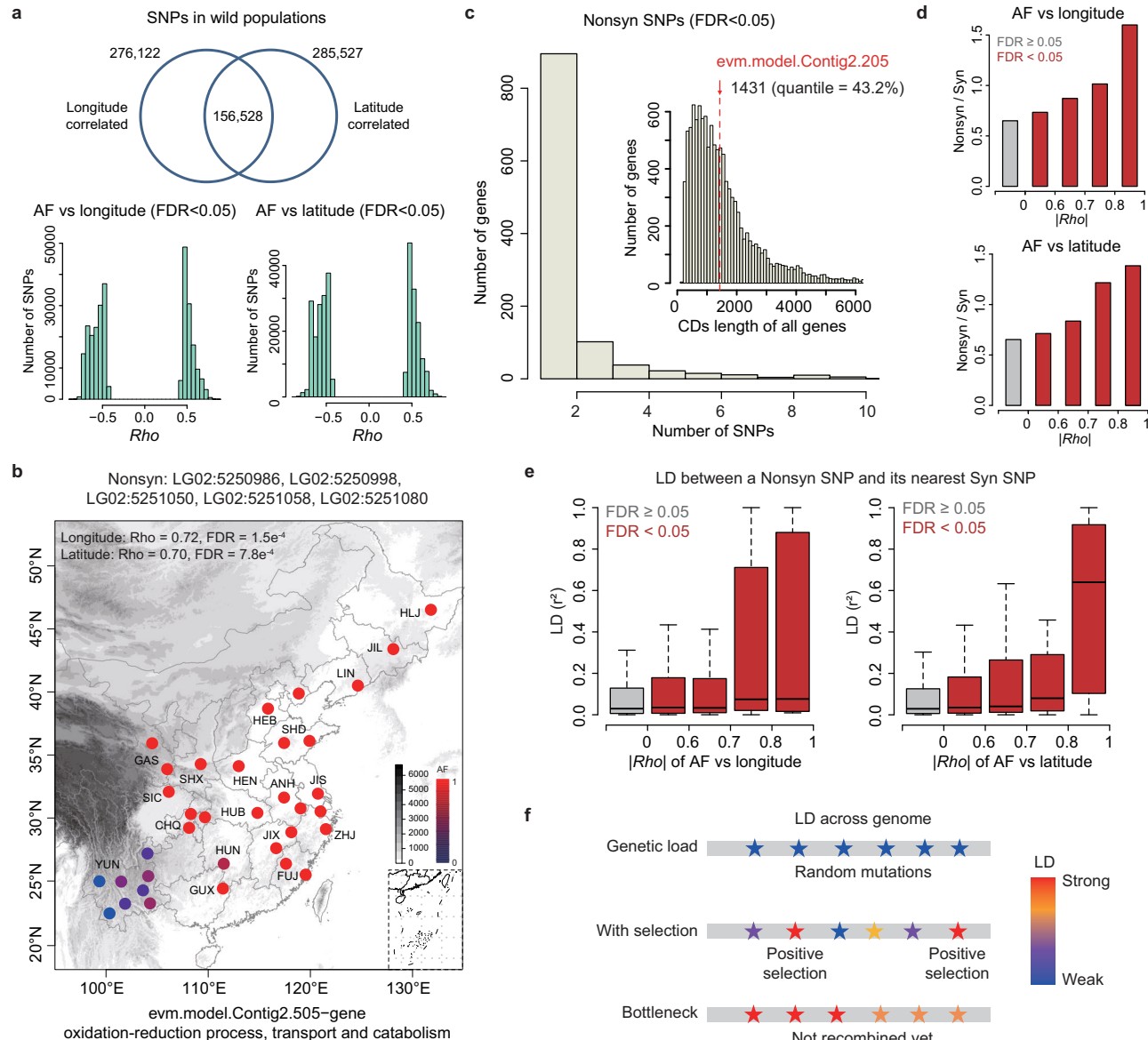

**Fig. 2 | Genetic evidence for environmental adaptation during the dispersal of *A. gifuensis* wild populations. a** Number of SNPs that show significant correlations (*FDR* < 0.05) between AF and the longitude/latitude of cities. The distributions of correlation coefficients (Spearman's *Rho*) are shown as histograms. **b** An example of nonsynonymous SNPs with significant longitude and latitude correlations. The map was drawn by the R Packages maps (https://cran.r-project.org/web/packages/maps). **c** Histogram showing the distribution of "the number of significant nonsynonymous SNPs" per gene. **d** Nonsyn/Syn ratios for SNPs with varying |*Rho*| values. Sites with *FDR* < 0.05 in the correlation test were in red. **e** Distribution of LD values between a nonsynonymous SNP and its nearest synonymous SNP. The X-axis

was divided with different ranges of |*Rho*| values. Numbers of sites are $n = 188{,}904$, 23,527, 6339, 2168, 13 for longitude and $n = 187{,}811$, 24,756, 7154, 1137, 93 for latitude. The range of boxes represents the 25%, 50%, and 75% quantiles and the whiskers represent the 2.5% and 97.5% quantiles. **f** LD landscape under different conditions. Genetic load accumulated from random drift should produce low LD. The selection on nonsynonymous SNPs will lead to stronger LD near the target sites compared with other genomic regions. Under bottleneck, the whole genome, rather than particular (scattered) regions, should exhibit strong LD. Source data are provided in the Source Data file.

(Supplementary Fig. 6). Similar genetic structure and diversity may indicate the successful establishment of artificially reared populations after release into the field.

Comparisons of runs of homozygosity (ROH) and inbreeding coefficient ($F_{ROH}$) with the three groups exhibited the same pattern: artificially bred and post-release groups presented longer ROH lengths, greater numbers of ROH segments, and higher inbreeding levels than their wild counterparts (Fig. 4b, c), although this pattern was not universal in each region (Supplementary Figs. 7 and 8). Linkage disequilibrium (LD) was measured using $r^2$ and the decay distance at half of the maximum $r^2$ value. For the three groups or each region, the

comparison of LD decay curves indicated that artificially bred and post-release populations had higher levels of LD, slower decay rates, and longer decay distances than their wild counterparts (Fig. 4d, Supplementary Fig. 9, and Supplementary Data 8).

It is known that generations of mass rearing has resulted in compromised parasitoid capacity and fecundity of *A. gifuensis*, such as reduced reproduction, decreased emergence rate and decreased adult longevity[36]. To investigate whether genetic changes have occurred in association with these phenomena, we compared artificially bred and wild populations from Yunnan Province. Based on $F_{ST}$, $\theta_\pi$ ratio and cross-population composite likelihood ratio (XP-CLR)

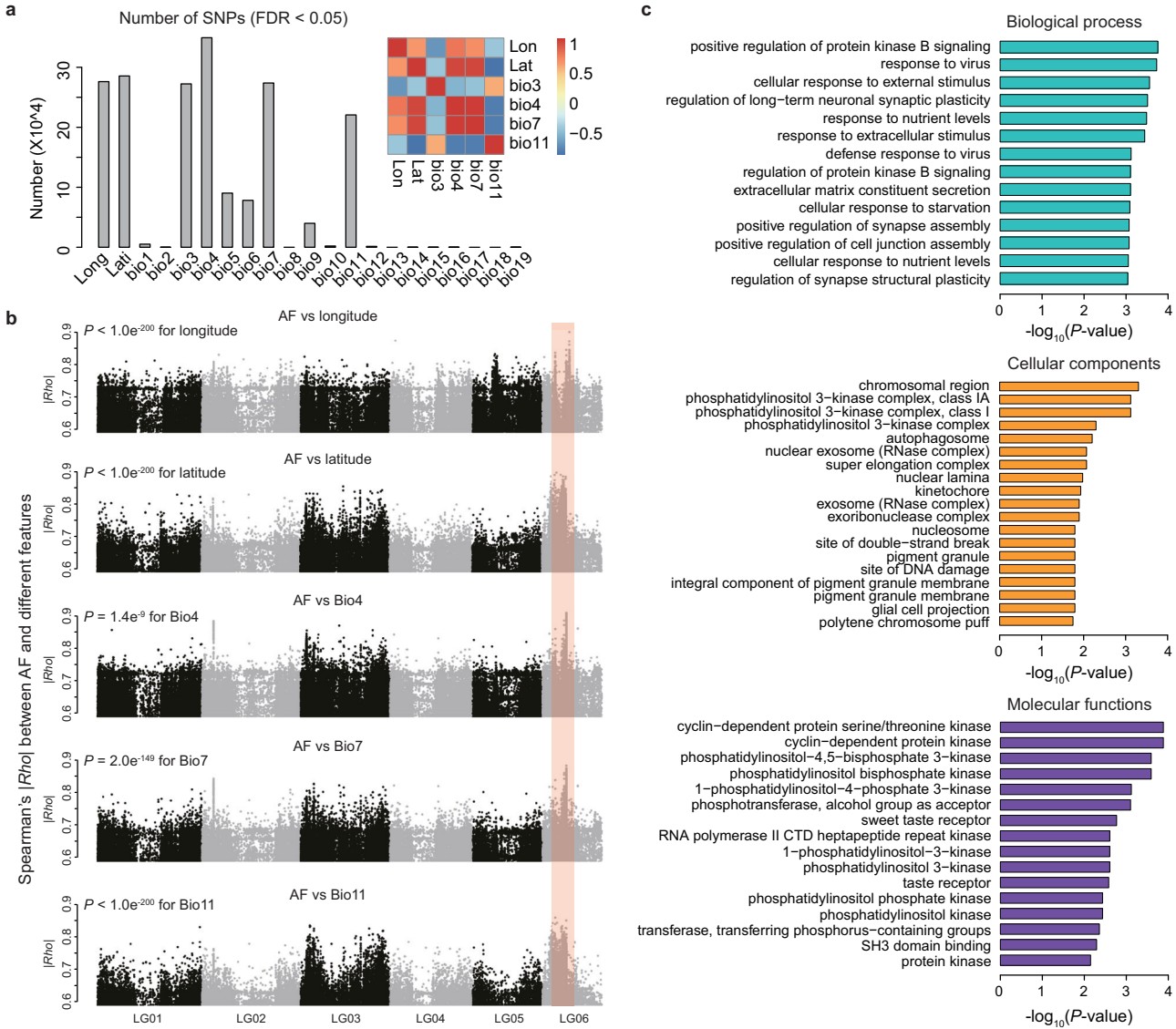

**Fig. 3 | Genes and genomic regions in wild *A. gifuensis* populations that are strongly correlated with geographic and climatic features. a** Number of SNPs that were significantly (*FDR* < 0.05) associated with different local climatic features. The features with more than $2 \times 10^5$ associated SNPs were retrieved. A heatmap of pairwise correlation was constructed for these features. Long: longitude; Lati: latitude; Bio3: Isothemality = mean diurnal range/Bio7; Bio4: Temperature seasonality (S.D.*100); Bio7: Annual temperature range (warmest - coldest); Bio11: Mean temperature of coldest quarter. **b** The absolute values of Spearman's *Rho*

between AF and various features were plotted along the *A. gifuensis* genome. The six chromosomes LG01 – LG06 are demonstrated. The orange rectangle denotes a merged region (8 Mb on chromosome LG06) containing multiple sites with |*Rho* | > 0.8. Unadjusted *P* values were calculated by two-sided Wilcoxon rank sum tests. Note that when *P* value is less than $1 \times 10^{-200}$, it will return *P* = 0 instead of an exact *P* value. **c** Gene ontology (GO) enrichment of all genes in the 8 Mb region on chromosome LG06. GO terms with *P* < 0.05 are shown. The unadjusted *P* values were used. Source data are provided in the Source Data file.

(Supplementary Fig. 10), 197 genes were identified as exhibiting differences between wild and artificially bred populations (Supplementary Data 9), thus maybe considered as candidate genes responsible for decreased parasitoid capacity and fecundity. Among these candidate genes, the enrichment of 20 related genes suggested different pathways of reaction to external signals, such as cellular response to external stimulus (GO:0071496), response to nutrient levels (GO:0031667, GO:0031669), response to extracellular stimulus (GO:0009991), and cellular response to starvation (GO:0009267) (Supplementary Data 10). Specifically, these 20 genes including phosphatidylinositol 4,5-bisphosphate 3-kinase (PI3K), SPARC, autophagy-related protein 5 (ATG5), cyclin-dependent kinase, gustatory receptors (GRs), and odorant receptor (OR) with the highest $F_{ST}$ value ($F_{ST}$ = 0.23), functioning in regulating lifespan, reproduction, or growth. There were three significant enrichments involved in cellular

component categories, chromosomal region (GO:0098687), and phosphatidylinositol kinase complex (GO:0005943 and GO:0097651). Notably, three selected genes enriched in cyclin-dependent protein kinase activity (GO:0004693) participated in the regulation of cell division or cell cycle[37–39]. Among the most enriched (*P* < 0.01) Kyoto Encyclopedia of Genes and Genomes (KEGG), AMPK signaling pathway and longevity regulating pathway were identified (Supplementary Data 11). These pathways play important roles in regulating growth and reprogramming metabolism[40]. In addition, we found that other proved lifespan-associated genes, such as serine/threonine-protein kinase[41], superoxide dismutase (SOD)[41], decaprenyl diphosphate synthases[42], mucin 5AC[43,44], peptidyl-alpha-hydroxyglycine alpha-amidating lyase 2[45], mitotic spindle assembly checkpoint protein (MAD1)[46], and transcriptional repressor protein YY1[47], were selected in artificially bred populations.

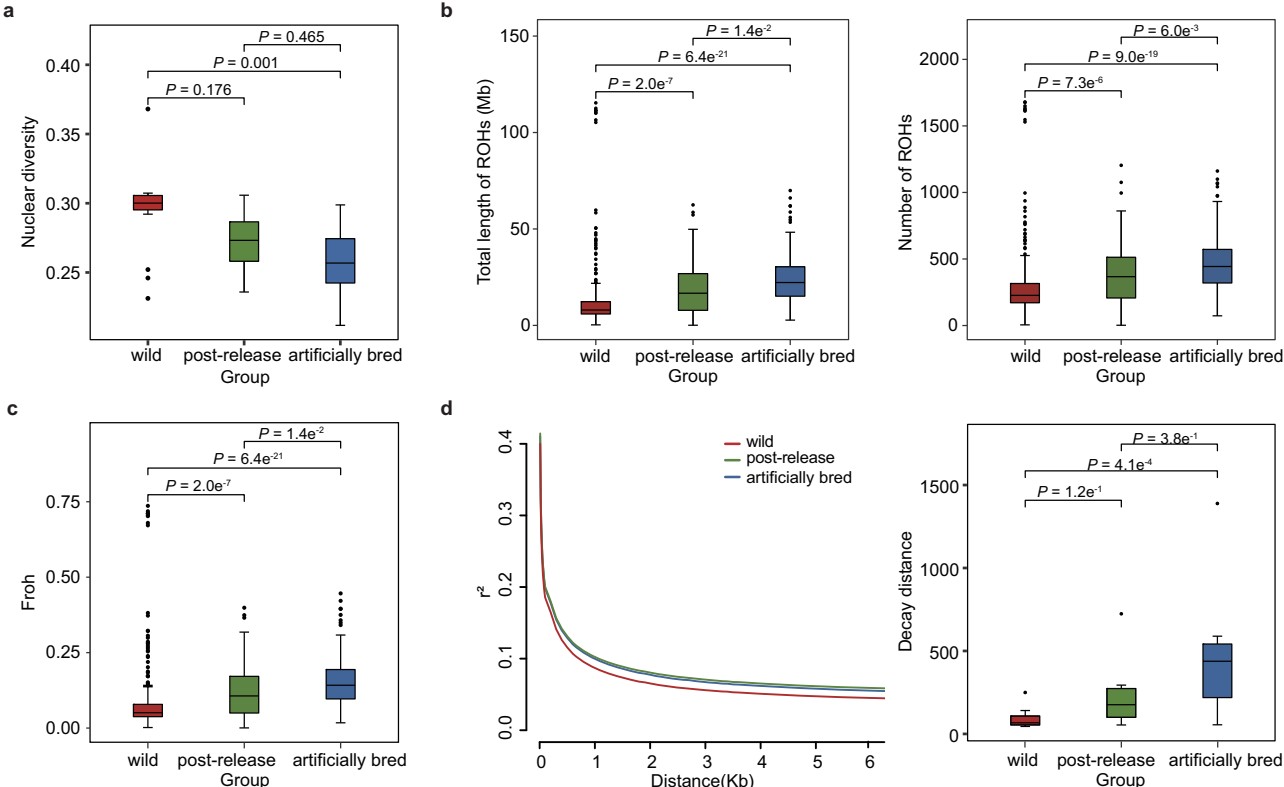

**Fig. 4 | Genetic differentiation emerged during long-term artificial rearing and biocontrol application. a** Comparison of nucleotide diversity between three distinct groups. The red, green, and blue boxes denote wild ($n = 17$), post-release ($n = 10$), and artificially bred populations ($n = 17$), respectively. **b** Quantity of the genome represented by runs of homozygosity (ROH) and the number of ROH segments in each group. **c** Inbreeding level among the three groups measured by ROH. The sample size of wild, post-release and artificially bred populations analyzed for ROH are 236, 117 and 160. **d** Decay of linkage disequilibrium in the wild (red), artificially bred (blue), and post-release populations (green). The boxplots depict the decay distance at half of the maximum $r^2$ value for each of the three groups. The number of wild, post-release and artificially bred populations analyzed for decay distance are 17, 7 and 14, excluding populations with less than 5 individuals. In all boxplots, significance is tested with two-sided Wilcoxon rank sum tests and $P$ value is adjusted via the Bonferroni test. Boxplots depict the median, 25%, 75% quartiles, and variance per group. The whiskers are 1.5*IQR (IQR: the interquartile range between the 25% and 75% percentile). Source data are provided in the Source Data file.

## Natural selection alleviates the post-release effects on wild populations

It is obvious that the wild and artificially bred *A. gifuensis* populations are facing completely different biotic and abiotic factors. For example, the bred populations were collected from mummied *M. persicae* feeding on tobaccos while the wild populations were collected from mummied *M. persicae* feeding on various host plants (Supplementary Data 1). Moreover, the natural environments are characterized by fluctuations and challenges, whereas the bred populations have been living in highly specialized environments or using other host aphid species[25] that might lead to genetic erosion as we observed. This raises a concern that the release of bred populations might pose a risk to the gene pool of wild populations, resulting in a reduction in their fitness and population size. Here, we asked whether various biotic or abiotic factors, like aphid host plants and environmental/climate conditions, could serve as selection agents to partially maintain the gene pool of wild populations.

To test whether the release of artificially bred populations (B) had detrimental effects on wild populations (N), we first conducted an ABBA-BABA analysis (*D*-statistics) implemented in Dsuite. Based on synonymous SNPs, significant genetic introgression between artificially bred or post-release populations and wild populations was found in Hebei (HEB), Henan (HEN) and Yunnan (Chuxiong, YUN01) ($P < 0.05$). By contrast, no significant genetic introgression was found based on missense (nonsynonymous) or all SNPs, except Yunnan (Chuxiong, YUN01) (Supplementary Data 12). As missense sites are

functionally more essential than synonymous sites, these results suggest that natural selection might have partially suppressed the impact of released population on wild populations. Then, we utilized alterations in the AF of SNPs (ΔAF) to measure their impact on a particular population. The following interpretation and analyses are described at province level. For wild populations (N) in a given province, the current AF was denoted as $AF_N$, and the "change in AF" ($\Delta AF_N$) was equal to $AF_N - AF_A$, where $AF_A$ represented ancestral allele frequency (Fig. 5a). For SNPs present in the wild population of a particular non-YUN province, the corresponding ancestral state $AF_A$ was inferred from YUN wild populations and outgroup *A. ervi*. For example, an SNP absent in YUN wild populations and *A. ervi* ($AF_A = 0$) but present in the current wild population of a non-YUN province ($AF_N > 0$) would indicate that the AF in the current non-YUN province N has been altered by $\Delta AF_N = AF_N - AF_A$ (Fig. 5a), presumably caused by one of the following three possibilities: (1) environmental adaptation during the dispersal of wild populations; (2) wild population affected by the locally released population; or (3) genetic drift. We have already identified SNPs associated with situation 1 (environmental adaptation) in previous sections. Therefore, the remaining SNPs under these criteria ($AF_A = 0$ & $AF_N > 0$, if any) would be associated with situation 2 or situation 3, which were irrelevant to local adaptation during spatial expansion. Under situation 2, $\Delta AF_N$ in a non-YUN province would represent the temporal evolutionary processes, as the YUN plus *A. ervi* information could represent the ancestral state (Fig. 5a). For situation 3 (random drift), our following analyses would disprove this possibility.

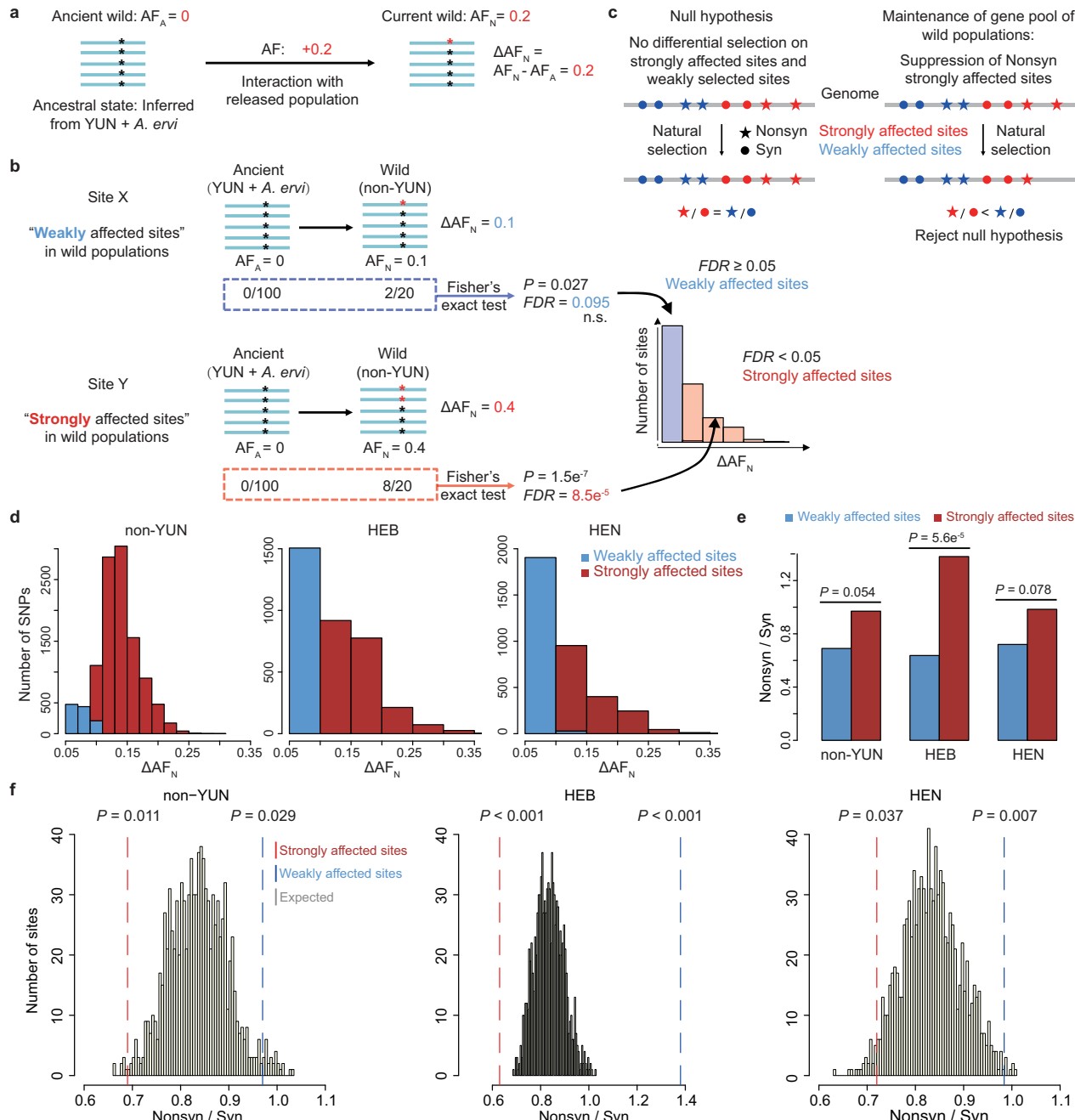

**Fig. 5 | The gene pool of the wild *A. gifuensis* population is maintained by natural selection. a** A schematic diagram illustrating the definition of ΔAF$_N$. The AFs of wild population were affected by the released population. The ancestral state of non-YUN wild populations could be inferred from the YUN population plus outgroup *A. ervi*. **b** Definition of strongly affected sites (red) and weakly affected sites (blue) according to the significance of ΔAF$_N$. Two-sided Fisher's exact tests were used and then adjusted by multiple testing correction to obtain an *FDR*. **c** Clarification of the hypothesis we are going to test based on the observed and expected Nonsyn/Syn ratios. **d** Histogram showing the numbers of SNPs of each category. HEB, HEN, or all non-YUN individuals were used. **e** Nonsynonymous to synonymous ratios of different sets of SNPs. Strongly affected sites (red) and weakly affected sites (blue) were compared. Unadjusted *P* values were obtained by two-sided Fisher's exact tests. **f** Observed and expected Nonsyn/Syn ratios for various sets of SNPs. HEB, HEN, or all non-YUN individuals were used. Bootstrap was done 1000 times. Unadjusted *P*-values were obtained by one-sided randomization tests. When none of the 1000 trials fall into the observed ranges between the two dashed lines, then *P* < 0.001 was recorded. Source data are provided in the Source Data file.

We aim to select the SNP sites meeting AF$_A$ = 0 < AF$_N$ as an example to prove that purifying selection would suppress the genetic (nonsynonymous) change in wild populations. This criterion (AF$_A$ = 0 < AF$_N$) was chosen owing to its explicit evidence for the derived mutations in the current wild populations. In some other cases, the trajectory of AF change is unclear and those sites are not suitable for inferring the effect of natural selection on AF, but the

constraint on mutations in wild populations still exists. As a result, the overall pattern is that in wild populations, AF$_N$ is stabilized by balancing selection so that the genetic diversity maintains highest in wild populations.

For SNPs meeting AF$_A$ = 0 < AF$_N$, where AF$_N$ is the AF of a non-YUN wild population, we try to quantify the effect of release on the wild population (N). We defined ΔAF$_N$ = AF$_N$-AF$_A$. Then, to measure the

extent of which the $AF_N$ has been affected by the released population, we first defined SNPs with $\Delta AF_N \geq 0.05$ as affected sites. Then we divided affected sites into "strongly affected sites" and "weakly affected sites" (Fig. 5b). Briefly, SNPs with significantly different AF between $AF_N$ and $AF_A$ were strongly affected sites and the remaining insignificant SNPs were weakly affected sites (Fig. 5b, see Methods for detail). Under random genetic drift (null hypothesis), strongly affected sites should show similar components (e.g., the Nonsyn/Syn ratio) with weakly affected sites (Fig. 5c). However, if natural selection maintains the gene pool in wild population ($AF_N$), then we expect that the strongly affected sites should show lower fraction of Nonsyn sites compared to the weakly affected sites due to the fact that some Nonsyn mutations should be eliminated by purifying selection before they could reach a remarkable change in allele frequency (Fig. 5c). Under this selection pressure, the gene pool in wild population would be saved.

Based on these notions, we set out to find suitable provinces to calculate $\Delta AF_N$ and define strongly and weakly affected sites. To ensure sufficient interaction between wild (N) and released (T) populations (where the released animals came from bred population B), we selected provinces with individual numbers $\geq 9$ for N, T, and B, which also minimizes the bias in AF introduced by insufficient individuals. Hebei (HEB) and Henan (HEN) provinces met these criteria. By requiring $AF_A = 0$ and $\Delta AF_N \geq 0.05$, we obtained 3523 and 3600 SNPs for HEB and HEN provinces, respectively. Among these SNPs, 2010 strongly affected sites and 1513 weakly affected sites were found in HEB, and 1666 strongly affected sites and 1934 weakly affected sites were found in HEN (Fig. 5d). Then, we calculated the Nonsyn/Syn ratios for different sets of sites. As expected, the strongly affected sites had remarkably lower Nonsyn/Syn ratios than the weakly affected sites (Fig. 5e). Moreover, our random shuffling results showed that the observed Nonsyn/Syn ratio for strongly affected sites was significantly lower than random expectation while the observed Nonsyn/Syn ratio for weakly affected sites was significantly higher than random expectation (Fig. 5f). These results suggest that the nonsynonymous mutations in the wild populations were largely suppressed owing to their deleterious effects. Although the released populations may affect the gene pool of wild populations, many functional sites (Nonsyn) in wild populations are maintained by natural selection.

To verify the robustness of our observations, we pooled all individuals from non-YUN provinces to calculate the respective AF values for N, T, and B. We also discovered that nonsynonymous changes were significantly underrepresented in the strongly affected sites of the wild non-YUN populations (Fig. 5d, e, f). This pattern is consistent with our observations in the HEB and HEN provinces, suggesting that natural selection has maintained functionally important sites in the wild populations, thereby constraining the effect of *A. gifuensis* release.

## Discussion

In this study, using *A. gifuensis* that has been utilized to control aphids in fields for decades, we analyzed population genetic diversity and evolutionary history of natively wild populations as the genetic background to determine the genetic and evolutionary consequences of human-mediated manipulation. Based on the mass rearing and biocontrol application history of *A. gifuensis* on *M. persicae* in China, we demonstrated its suitability and safety by highlighting the divergence between the wild and artificially bred populations, as well as the limited effects of release into wild fields. Meanwhile, we acknowledge that a comprehensive assessment of the impact of releasing this parasitoid wasp will necessitate investigations involving populations using other aphid hosts. This generalization requires empirical evidence from bioassays to demonstrate that parasitoid performance is not significantly affected by introgressions into wild populations and populations using other aphid hosts.

Our results revealed that populations from Yunnan represent an ancestral state and are genetically distinct from other peripheral populations. By examining the correlation between the variations in each population and local geographic features such as longitude, latitude, and bioclimatic variables, we discovered that excessive nonsynonymous mutations (compared to synonymous mutations) were associated with local geographic features and located in evolutionarily conserved genes, indicating that these nonsynonymous mutations were likely involved in the environmental adaptation during the dispersal of wild *A. gifuensis* populations. Meanwhile, these conclusions are totally based on in silico data and more delicate measurements should be designed to robustly confirm. Since the historical data of these populations were not available, we acknowledge that our inference of adaptation might be complicated by other unknown factors although we have tried to conceive several methodologies to distinguish between different possibilities. Nevertheless, we believe that this is an essential step towards the identification of the genomic relic of natural selection on environmental adaptation using the large-scale population data. The population genetic system of wild *A. gifuensis* helps us understand the genetic diversity pattern during rearing and post-release, and might provide an opportunity for measuring the effect of long-term mass rearing and releases on wild populations.

Comparisons between human-intervened populations (artificially bred and post-release populations) and wild populations revealed that human-intervened populations exhibited different patterns of population genetic differentiation, lower nucleotide diversity, and higher inbreeding level than wild populations. These results were consistent with the general pattern of reduced genetic diversity during long-term breeding[48,49] and environmental adaptation during rearing. Commercialized *A. gifuensis* populations were established from a small wild population, which likely led to extensive inbreeding over generations of artificial rearing. Under genetic drift, extensive inbreeding causes individuals within artificially bred populations to become more genetically similar to one another compared to wild populations[50,51], as a result of distinct genetic compositions of artificially bred *A. gifuensis* populations. However, the similar genetic structure and diversity of the artificially bred and post-release populations in each region indicated the successful establishment of artificially reared populations after release in fields. Additionally, a higher level of LD has been found during the domestication process of economically important animals and plants[52,53]. For the parasitoid wasp *A. gifuensis*, despite the frequent introduction of wild populations to expand the artificially bred populations, the latter still exhibited stronger LD compared to the former. The introduction and release application of *A. gifuensis* can be complex, primarily because not all captive populations had the same origin (Supplementary Data 2). The diversity in origins may lead to differences in population genetic diversity and inbreeding level among artificially bred populations, necessitating more meticulous sampling and monitoring after the release of captive *A. gifuensis* in each region.

Long-term artificial rearing satisfies the requirements for suppressing *M. persicae* populations but suffers from rapid population decline and a substantial reduction in fecundity or parasitoid ability[36]. Based on our extensive collections and genomic resequencing of samples from Yunnan Province, which has experienced a prolonged rearing and biocontrol history of *A. gifuensis*, we were able to study the genetic consequences of long-term mass rearing. It was predicted that the majority of the 197 candidate genes, which exhibited differences between wild and artificially bred populations, play important roles in regulating growth and reproduction. PI3K is a key component of the insulin/insulin growth factor I signaling pathway, which regulates lifespan, reproduction, and growth[54]. SPARC has also been proven to affect oogenesis and embryo development in insects[55] and is essential for maintaining tissue integrity[56]. ATG5 induces apoptosis following autophagy[57] and is highly transcribed during metamorphosis in most insects[58-60]. The lifespan-associated genes serine/threonine-protein

kinase and SOD were identified in *Pteromalus puparum*[41] and were enriched in longevity regulating pathway. OR, OBP, and IR are odor-related proteins for insects with the function of specific odor selection and peripheral signal transduction, which may be associated with the host-finding behavior and host fidelity of parasitoid wasps[61]. Four of the five identified GRs enriched in the response pathway to nutrient levels were associated with sugary taste. In *Drosophila*, sugar receptors contribute to feeding behavior[62], which may be caused by different feeding conditions between the rearing environment and the natural environment. Notably, genes involved in chemical perception may be associated with loss of host fidelity (i.e., detection of chemical cues during the parasitoid development and adult emergence). Such as *A. ervi*, inbreeding frequently could induce its loss of host fidelity thus influence the host finding in heterogeneous environments, and the trait of host fidelity could be learned from chemical signals emitted from the host in which parasitoids developed[12]. To determine whether a relationship exists between these chemical perception genes and the host fidelity, we conducted GWAS using individuals collected from different host plants (see Methods for details). Pairwise comparisons were performed between four host plants *Nicotiana*, *Brassica*, *Capsicum*, and *Solanum*. We selected the top 5% of the most significant sites of each comparison, but no chemical perception genes were found among the host genes of the SNPs (Supplementary Fig. 11 and Supplementary Data 13). This suggests that in our cases, the chemical perception genes identified in the selective sweep analyses may not represent the most critical (top 5%) determinants of host fidelity, or some unknown technical issues limited the detection power. On the other hand, it was difficult to quantitatively estimate effects of the identified chemoreceptor genes on its host fidelity. However, we still reserve the possibility that the chemical perception genes might partially contribute to the adaptation to different aphid host plants. We also identified a detoxifying gene, ABC transporter G family (ABCG), which plays a crucial role in the transport of various substances, including lipids and toxins. The ABCG has been demonstrated to be involved in insecticide detoxification in insects, such as *Bemisia tabaci*[63], *Nilaparvata lugens*[64], *Sogatella furcifera*[65], *Anopheles gambiae*[66], *Anopheles stephensi*[67], *Tribolium castaneum*[68]. Due to the lack of chemical control records in the fields, it remains challenging to determine whether the different diversity of ABCG gene between *A. gifuensis* wild and artificially bred populations was associated with insecticide resistance. When compared to wild populations, these candidate genes may serve as indicators of genetic erosion resulting from long-term mass rearing, and changes in the expression of these genes may affect the behaviors or physical fitness of *A. gifuensis*.

The aforementioned symptoms of decreased population diversity or variations, and the differentiated genes identified under the comparison between artificially bred and wild populations of Yunnan, reflected genetic erosion during the long-term captivity of *A. gifuensis*. Genetic erosion is a cause of endangerment of small isolated populations and may affect their survival performance[69,70]. Hence, it would be prudent to manage the genetic diversity of captive parasitoids before release into fields to mitigate the adverse effects of prolonged inbreeding through the genetic rescue method, incorporating natural populations with higher genetic variation[12,71]. If introducing intraspecific hybridization by mixing different populations or mitochondrial haplotypes was proven to be effective, the wild population, especially the Yunnan populations, should be considered an important resource to promote genetic improvement according to our assessment. In addition, an effective genomic improvement approach should be investigated for ongoing rearing and application of this biocontrol agent.

Population genetic divergence was greatest between the wild and artificially bred groups. In conjunction with the results of the comparison of genetic diversity, which showed wild > post-release > artificially bred, it indicated that the release of breeding populations does not presently have a destructive effect on the genetic diversity of wild populations. However, there are several factors that may influence the parasitoid behavior, consequently influence the genetic variations and population structure of parasitoid wasps, such as host species[72], sex ratio[14], hyperparasitoids[73], symbiont diversity[74,75], insecticides[76] or climate change[77]. To investigate the factors influencing genetic differentiations between wild and artificially bred populations, we collected female adults from the same host aphid, *M. persicae*, and conducted the genetic comparisons between populations from the same region or with the same population structure. Under this strategy, we supposed natural selection would maintain the genetic diversity of wild populations during long-term release application of captive *A. gifuensis*.

To quantitatively test our hypothesis, we inferred the ancestral state of the YUN population and demonstrated that strongly affected genomic regions in non-YUN wild populations avoided functionally important sites or genes, suggesting that natural selection maintained the gene pool of wild populations. Although the released populations affected the local wild populations to some extent, it is important to note that the extant YUN populations continued to exhibit the highest genetic diversity and largely maintained their ancestral state. Therefore, even if some local populations are strongly affected by the release, in theory, their diversity might be restored through the introduction of YUN populations.

As our null hypothesis proposed, the differential performance between nonsynonymous and synonymous SNPs might serve as evidence indicating that our observation is not merely random, and natural selection is likely to participate in this process to maintain the frequency spectrum of nonsynonymous sites. The selection agents are likely to be many biotic and abiotic factors. Bred populations parasitizing *M. persicae* aphids on tobacco and wild populations fed on diverse plants. For examples, all HEN wild individuals were collected from aphid mummies on *Brassica napus* and >70% of the HEB wild individuals were collected from aphid mummies on *Lycopersicon esculentum* (Supplementary Data 1). These factors might exert selection pressure on maintaining the gene pool of wild populations. However, the lack of historical data, such as the genomic information of wild populations prior to the application of biocontrol agents, hampered the quantitative assessment of the evolutionary consequences of release. Thus, our analyses were not the direct inspection of the temporal evolutionary process, and there was the possibility that some alternative explanations accounting for our observations. Deeper works in the future are needed to give a more solid and confirmative conclusion. Nevertheless, the data presented in this study can be utilized as long-term monitoring resources for assessing ecological security for future exploitation.

Most of our samples were collected from Yunnan Province. Although the YUN populations had the highest genetic diversity and were believed to be the ancestors of all populations in China, the limited sample size from non-YUN regions could have affected the estimation of population history, environmental adaptation, or other genetic parameters. Additional samples from non-YUN populations would help to increase the statistical power and confidence of our analyses. Additionally, we used the YUN populations to infer the ancestral state of the genome sequences of the non-YUN population. Future study involving, if large-scale time-course or historical data would help to directly dissect the shift in genetic architecture during long-term rearing and quantify the effect of release on wild populations.

## Methods
### Sampling collection
We collected *A. gifuensis* adults from mummified *Myzus persicae*. Due to the haplodiploid sex determination system of *A. gifuensis*, only diploid females were selected for resequencing. To investigate whether the continuous release of mass-reared individuals affected

wild individuals, we collected both wild populations and human-intervened populations. For wild populations, 265 wild individuals were collected far away from areas where *A. gifuensis* was used for biocontrol (Supplementary Data 1), considering the flight distance of mummified alates of *M. persicae* (-2.16 km) as an indicator of parasitoids dispersal[78]. Among the collected wild populations, the distance between wild and post-release populations was from 186.27 km to 11,807.19 km, which calculated based on longitude and latitude. For human-intervened populations, we firstly collected 160 intensively captive individuals from 15 laboratories or factories where artificially rearing has been carried out for over ten years (Supplementary Data 2). Additionally, 117 individuals were collected from experimental tobacco fields after release during the period between June and August (Supplementary Data 2). Typically, in tobacco fields, two or three releases of *A. gifuensis* were conducted at different growth stages of tobacco, involving approximately 10,000–15,000 individuals per hectare for each release[26]. Accordingly, we referred to the three hierarchical groups as wild populations (N), artificially bred populations (B) and post-release populations (T). More detailed collection information of wild populations and artificial rearing records of the samples are listed in Supplementary Data 1 and Supplementary Data 2.

### DNA extraction and sequencing
The samples were stored in absolute ethanol at −80 °C for subsequent DNA extraction. Each sample was firstly identified as member of *Aphidius* based on morphological characters[79], and combined with DNA barcoding sequences (partial mitochondrial *cox1* gene) to identified species[80]. Genomic DNA was extracted from each individual using the DNeasy Blood and Tissue Kit (Qiagen, Germany) following the standard protocol and quantified using 0.8% agarose gel electrophoresis and Qubit 3.0 fluorometry (Invitrogen, USA). Genomic sequencing libraries with an insert size of 300–600 bp were constructed, following the Illumina library preparation instructions, and sequenced on the Illumina NovaSeq platform (Illumina, USA) with paired-end (PE) 150 bp reads.

### Data quality control, genome mapping, and variant calling
The sequencing quality control of raw data was performed using fastp 0.21.0[81] with default parameters to eliminate low-quality reads and adapter sequences. The clean data were mapped to our *A. gifuensis* reference genome (GenBank: GCA_014905175.1)[82] using BWA-MEM v0.7.12-r1039[83] with default parameters, and samples with low mapping rates (< 80%), average sequencing depth (< 10×) and coverage (< 80%) were removed (Supplementary Data 3). Mapping results of each individual sample were sorted and converted to BAM format using SAMtools v1.9[84]. Picard v2.21.6 (http://broadinstitute.github.io/picard/) was used to remove duplicate reads.

After mapping, we performed SNP calling using the Genome Analysis Toolkit (GATK) v4.1.9.0[85]. The HaplotypeCaller strategy was used to identify variants in each sample. All GVCFs of each sample were merged for joint genotyping to generate numerous variants. Then, we split and extracted SNPs and indels. Raw SNPs were filtered using VariantFiltration with the parameters "QUAL < 30.0 | QD < 13.0 | MQ < 20.0 | FS > 20.0 | MQRanKSum < -3.0 | ReadPosRankSum < -3.0". Using VCFtools v0.1.16[86], only high-quality biallelic SNPs were retained for subsequent analysis, with missing rates ≤ 0.15 (--max-missing 0.85), minor allele frequencies ≥ 0.05 (--maf 0.05), and mean depth ≥ 10 (--min-meanDP 10). SNPs were further annotated using SnpEff v4.3t[87] based on the annotation information of the *A. gifuensis* reference genome.

### Mitochondrial genome assembly
We mapped the clean resequencing data of each individual to the reference mitogenome of *A. gifuensis* (GenBank: MT264907.1)[80] to generate novel mitogenomes using Geneious v11.1.4 (http://www.geneious.com/), with a maximum mismatch of 5%, a maximum gap size of 5 bp, and a minimum overlap of 40 bp with at least 95% identity. Each novel mitogenome was annotated using the reference sequence.

### Population genomic analyses
To better reflect the genomic diversity and structure among populations, we firstly calculated the kinship coefficient between individual pairs of intro-populations using KING v2.2.5[88], leaving only one individual to represent the group with the most closely related pedigree. According to the protocol regarding the inference criteria based on estimated kinship coefficients ($\phi$) for different kinship degrees, ranges of > 0.354, [0.177, 0.354], and [0.0884, 0.177] correspond to duplicate/monozygotic twin ($\phi = 1/2$), 1st-degree ($\phi = 1/4$), and 2nd-degree ($\phi = 1/8$) relationships, respectively. The kinship coefficient between individuals estimated to exceed 0.125 was considered to be from the same pedigree, and 28 individuals were ultimately removed (Supplementary Data 1 and 4).

Genetic diversity was calculated using the *populations* function of Stacks v2.60[89,90], including nucleotide diversity ($\theta_\pi$), excepted heterozygosity ($H_e$), observed heterozygosity ($H_o$). Weir and Cockerham's estimator of $F_{ST}$ was calculated using the R package SNPRelate[91] to measure the genetic differentiation among multiple populations. Meanwhile, we utilized VCFtools to estimate the pairwise-$F_{ST}$ among the artificially bred, post-release and wild groups. We used the formula $Nm = (1- F_{ST})/4 \cdot F_{ST}$ to analyze the gene flow levels between the wild populations.

To avoid the influence of LD on the population genetic structure, the data set was thinned to one SNP per 3k window to generate independent loci and pruned with the parameter --indep-pairwise (50 10 0.2) using PLINK v1.90b6.18[92]. Population structure analysis with independent SNPs was conducted with ADMIXTURE v1.3.0[93], and the number of hypothetical ancestral clusters K ranged from 2 to 10 (Supplementary Fig. 2). K value with the lowest cross-validation error was chosen for analyses. PCA was performed using PLINK with the same dataset. Moreover, BAPS v6.0[94] was utilized to predict the clustering of individuals with mitochondrial genomes by combining the sample locations with the likelihood of data.

### Phylogenomic reconstruction
Phylogenetic relationships were estimated using nucleic SNPs and mitogenome datasets. To ascertain the original of wild *A. gifuensis*, we chose *A. ervi* as an outgroup. Sequencing reads of *A. ervi* were obtained from the Bioinformatic Platform for Agrosystem Arthropods (https://bipaa.genouest.org/sp/aphidius_ervi/download/genome/v3.0/) and aligned with *A. gifuensis* genome using BWA-MEM v0.7.12-r1039[83], following the same method as described above. Based on the nucleic SNPs dataset of *A. ervi* and *A. gifuensis* wild populations, a ML phylogenetic tree was reconstructed using RAxML v8.2.12[95] under the GTR + GAMMA model. The protein-coding genes (PCGs) of mitogenomes were exported and haplotyped with DnaSP 6.0[96], and an ML phylogenetic tree with 1000 bootstraps was reconstructed using IQ-TREE 1.6.5[97].

### Population demographic history inference
Individual sequencing depths greater than 20× of wild populations were selected to estimate the effective population size ($N_e$) of *A. gifuensis* over time, using SMC++[98] with higher resolution for recent demographic history than other methods such as PSMC or MSMC. We used a mutation rate of $3.4 \times 10^{-9}$ per base pair as estimated for the bumblebee *Bombus terrestris*[99] and approximately 15 generations of *A. gifuensis* per year[100,101].

### Linkage disequilibrium (LD) and estimation of inbreeding
To measure the degree of LD among multiple groups, including artificially bred, post-release, and wild populations, we calculated the

correlation coefficient ($r^2$) based on the increasing decay distance and plotted LD decay curves using PopLDdecay[102], using all high-quality biallelic SNPs that were filtered. The LD ($r^2$) between a nonsynonymous SNP and its nearest synonymous SNP was calculated with similar strategy. Given two SNP sites, the numbers (allele counts) of four potential haplotypes were extracted to calculate the four haplotype frequencies g11, g10, g01, and g00 as defined by Lewontin 1988[103]. Then, the allele frequencies $P_1$, $Q_1$, $P_2$, $Q_2$, and the LD parameters $D$, $X^2$, and $r^2$ were calculated according to the original study Lewontin 1988[103].

We measured ROH per individual using PLINK[104] to assess the potential impact of long-term mass rearing and biocontrol application of *A. gifuensis* on inbreeding. The ROH was calculated with the parameters "--homozyg-window-snp 50 --homozyg-density 50 --homozyg-window-het 1 --homozyg-gap 100 --homozyg-kb 20 --homozyg-snp 50." Genome inbreeding coefficient ($F_{ROH} = \sum L_{ROHk}/L$) was used to assess the inbreeding level of each individual of each group, where ROH represented runs of homozygosity, $L_{ROHk}$ denoted the length of the *kth* ROH, and L indicated the genome size.

### Detection of selective sweeps

Selective sweeps of *A. gifuensis* were based on filtered nucleotide SNPs using three approaches: genome-wide distribution of population fixation statistics $F_{ST}$[105], nucleotide diversity $\theta_\pi$[106], and the XP-CLR approach[107]. We selected the dataset of the Yunnan populations, which have the longest rearing and biocontrol history of *A. gifuensis*, in order to detect genomic selection signals. We utilized VCFtools to calculate $F_{ST}$ and $\theta_\pi$ with a sliding window size of 10 Kb and a step size of 5 Kb. Negative $F_{ST}$ values were considered to be 0. For the results of $\theta_\pi$, we further calculated the $\log_2(\theta_{\pi\text{-artificially bred}}/\theta_{\pi\text{-wild}})$. The XP-CLR approach was implemented using a Python module with a sliding window size of 10 Kb, a step size of 5 Kb, and no more than 600 SNPs per window. Finally, the high-divergence regions resulting from the intersection of the top 5% $F_{ST}$, $\log_2(\theta_{\pi\text{-artificially bred}}/\theta_{\pi\text{-wild}})$, and XP-CLR values were subsequently deemed to be putatively selective sweeps.

Genes from overlapping candidate regions were annotated based on the reference genome of *A. gifuensis*. GO enrichment and KEGG pathway analyses were performed using the R package clusterProfiler[108]. Only results with $P < 0.05$ were considered.

### Genome-wide association studies (GWAS) based on host plant species

The wild populations collected from different host plant species were used for genome-wild association studies (GWAS), including *Nicotiana*, *Brassica*, *Capsicum* and *Solanum*. The Efficient Mixed-Model Association eXpedited (EMMAX) was used to analyze the relative kinship of theses populations[109]. The significant $P$-value thresholds of the GWAS were set as $-\log_{10}P$ ($P = 1/\text{total SNPs}$). The quantile-quantile (Q-Q) plot was shown with the expected $P$ value and $-\log_{10}P$ of each SNP, and the Manhattan plot was demonstrated using the R package qqman[110].

### Gene flow inferring (*D*-statistics and ABBA-BABA)

The ABBA-BABA analysis was performed on a typical four-taxon tree (((P1, P2), P3), O), where P1 and P2 are recognized to be closely related, P3 is a third ingroup taxa and O represents an outgroup served to define the ancestral allele. To test the genetic introgression between post-release or artificially bred populations and wild populations, ABBA-BABA test (*D*-statistics) was conducted with Dsuite v0.5[111] using *A. ervi* as an outgroup. We designed artificially bred, post-release and wild population province as P1, P2 and P3 for each province. The analyses were conducted based on missense, synonymous and non-coding SNPs, respectively. A significant deviation of *D*-statistics from zero indicates the occurrence of gene flow between P3 and P1 (D < 0) or between P3 and P2 (D > 0). A combined result with $P < 0.05$ and $|Z$-scores| > 3 was taken as significant evidence of introgression.

### Allele frequency (AF) analyses

To identify SNPs associated with environmental adaptation, we used the 35 wild populations (N). For each SNP site, we calculated the AF among wild individuals collected in each city. We have polarized the direction of the mutation according to the outgroup species *A. ervi*. For a particular bi-allelic SNP site in the *A. gifuensis* population, either the reference allele or the alternative allele is identical to the *A. ervi* sequence (Supplementary Fig. 12a). In the former case, the reference allele of *A. gifuensis* should be the ancestral allele, then derived allele frequency (DAF) = alternative allele frequency AF. In the latter case, the alternative allele in *A. gifuensis* is likely to be the ancestral allele and thus DAF = 1 − AF (Supplementary Fig. 12a). In brief, when the reference allele was the derived allele and the alternative allele was identical to the sequence in outgroup *A. ervi*, we changed the AF to 1-AF for all populations (provinces). We calculated Spearman's correlation (*Rho*) between the AFs among various cities and their corresponding longitude or latitude. *P*-values were adjusted for multiple testing corrections. *FDR* < 0.05 was regarded as statistically significant.

In contrast, when evaluating the effect of the release on wild populations, only the non-YUN provinces were investigated. We used the alteration in allele frequency (ΔAF) to quantify the impact on a particular population. For *A. gifuensis* wild populations (N) in a particular non-YUN province, the present AF was denoted as $AF_N$, and the "change in AF" = $\Delta AF_N = AF_N\text{-}AF_A$, where $AF_A$ was the AF in ancestral population inferred from YUN population plus outgroup species *A. ervi*. The AF in a specific non-YUN wild population ($AF_N$) is exact the same as we previously defined (Supplementary Fig. 12a). However, to ensure that the $AF_N$ truly represents ΔAF, we selected the sites with the ancestral $AF_A = 0$. It means that all the alleles in YUN wild individuals and the *A. ervi* sequence were identical, thus representing the ancestral allele (Supplementary Fig. 12b). In other words, for this analysis, not all SNPs in a non-YUN province were used; rather, the AFs of the specific SNPs of interest were exactly identical to those defined in the correlation test (Supplementary Fig. 12a). The rationale for exclusively focusing on SNPs with ancestral $AF_A = 0$ was that in other cases, it becomes challenging to ascertain the $AF_A$, making it difficult to determine the change in $AF_N$. However, in the correlation tests between SNPs and regional factors, we do not interrogate the change in AF within a specific time scale, eliminating computational challenge in this regard. For HEB and HEN provinces with presumably sufficient interaction between wild and released population, we selected the SNPs with $AF_A = 0 < AF_N$ and let $\Delta AF_N = AF_N\text{-}AF_A$. SNPs with $\Delta AF_N \geq 0.05$ were defined as affected sites. Among these affected sites, SNPs with significantly different AF between $AF_N$ and $AF_A$ were strongly affected sites and the remaining insignificant SNPs were weakly affected sites. Significance was defined by Fisher's exact test on reference and alternative allele counts. $P$ value was then adjusted by multiple testing correction to obtain an FDR. SNPs with FDR < 0.05 was defined as "strongly affected sites".

During the calculation of the expected Nonsyn/Syn ratios, the $AF_A$ and $AF_N$ were randomized for 1000 times for each province (HEB or HEN), respectively. Subsequently, $\Delta AF_N = AF_N$ - $AF_A$ was calculated and the "strongly affected sites" and "weakly affected sites" were defined. The expected Nonsyn/Syn ratio was calculated accordingly. Each round of randomization in each province will produce one Nonsyn/Syn ratio. For each province of HEB, HEN, or pooled non-YUN, the randomization was done for 1000 times.

### Calculation of nonsynonymous and synonymous substitution rates (dN, dS) of coding genes

To measure the conservation levels of genes in *A. gifuensis*, we calculated the sequence divergence *dN/dS* ratio in comparison to *A. ervi* genome (Bioinformatic Platform for Agrosystem Arthropods: https://bipaa.genouest.org/sp/aphidius_ervi/download/genome/v3.0/). We aligned the longest predicted proteins of single-copy orthologs using

MUSCLE v3.8.31[112] and transformed amino acid alignments to codon-based nucleotide alignments using Pal2Nal[113]. The pairwise *dN/dS* ratio of each gene was calculated using the yn00 algorithm in PAML[114].

## Reporting summary
Further information on research design is available in the Nature Portfolio Reporting Summary linked to this article.

## Data availability
The raw sequencing data from this study can be found in the Sequence Read Archive (SRA) under Bioproject accession number PRJNA852353, with the accession numbers of SRR19960681–SRR19961222 (Supplementary Data 14). All mitogenomes have been deposited in GenBank under the accession numbers OP002317–OP002856 (Supplementary Data 15). Reference genome of *Aphidius gifuensis* (GenBank: GCA_014905175.1) and its mitogenome (GenBank: MT264907.1) can be found in NCBI. Source data are provided with this paper which are deposited in Figshare database (https://doi.org/10.6084/m9.figshare.25038722).

## Code availability
All code and software sources used in our paper are listed in the "Methods" section with corresponding citations of references and Source Data file.

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

## Acknowledgements

This study was supported by grants from the National Natural Science Foundation of China (No. 31922012), the China National Tobacco Corporation of Science and Technology Major Projects (No. 110202001036 [LS-05]), the International Postdoctoral Exchange Fellowship Program from the China Postdoctoral Council (PC2021094), the 2115 Talent Development Program of China Agricultural University, the project of the Northeast Asia Biodiversity Research Center (2572022DS09) and Expert Workstation in Zhaotong, Yunnan (Nos. 2019ZTYX03, 2021ZTYX05). We thank Yunfei Wu, Zhuo Chen, Qian Zhao, Qiaoqiao Liu and Xinzhi Liu for help with the sample collection.

## Author contributions

H.L., Z.L., W.C. and H.Y. designed the research. S.L., L.T. and F.S. provide constructive advices for analyses. B.L., X.W. and Z.F. prepared samples and performed experiments. Z.D. and Z.F. performed mtDNA analyses. B.L. and Y.D. performed genome data analyses, wrote and revised the manuscript with comments from the other authors.

## Competing interests

Z.L. is a full-time employee of Yunnan Tobacco Company and H.Y. is a full-time employee of Tobacco Company of Yuxi. These two companies are the subordinate units of China National Tobacco Corporation. The remaining authors declare no competing interests.
