## [Peer Review File · Nature Communications]

Natural selection and genetic diversity maintenance in a parasitic wasp during continuous biological control applicationReviewers' Comments:

Reviewer #1:

Remarks to the Author:

In this manuscript, Bingyan Li et al. report a population genomic survey of the aphid parasitoid *Aphidius gifuensis* in China to determine whether the release of mass-reared parasitoid wasps may alter the genetic features of natural populations of the same species. The authors are particularly interested in studying the effects of inbreeding on parasitoid performance at the field scale, and they looked for genomic evidence to support that idea. The work included the sampling of wild and cage-reared populations, also including samples of parasitoids from an aphid population that is currently under biological control. The authors generated a massive amount of data from genome resequencing. Then they described the data with no clear hypothesis other than stating that natural selection maintains genetic diversity in wild populations. However, they don't discuss what selection agents are putatively acting on populations and their significance. Hence, my main concern is related to data interpretation and discussion, as in its present status, the manuscript is too descriptive, not going into details about how the biology of the parasitoid and the role of its interactions with other organisms and the environment can explain the genetic diversity, as I review below. Consequently, based on what I read, I do not recommend this manuscript to be accepted in its present form, as much work is needed to be of wide interest to the readers of Nature Communications.

Lines 34-35: The statement is not entirely accurate as there are examples of successful biological control programs of aphids based on single introduction events with no further releases of biocontrol agents (e.g., parasitoid wasps to control cereal aphids in South America), as reviewed in: Hance, T., Kohandani-Tafresh, F., and Munaut, F. (2017). Biological control. In *Aphids as crop pests*, H. Van Emden and R. Harrington, eds (CABI Publishing), pp. 448–493.

Lines 37-40: There are examples of aphid parasitoids losing specialization due to inbreeding. For instance, host fidelity (i.e., parasitoids prefer the natal aphid–host–plant system from where they emerged and increase their foraging efficiency) can be lost due to inbreeding, as reported in: Sepúlveda, D. A., Zepeda-Paulo, F., Ramírez, C. C., Lavandero, B., & Figueroa, C. C. (2017). Loss of host fidelity in highly inbred populations of the parasitoid wasp *Aphidius ervi* (Hymenoptera: Braconidae). *Journal of Pest Science*, 90, 649-658.

Line 44: You mean balancing selection? What selection agents may be involved? Explain a bit more about how and under what circumstances natural selection can preserve the genetic diversity of a population.

Line 47: Replace abnormal with increased.

Lines 51-52: Where is this parasitoid species originated? I assume the species is endemic to East Asia, and that's why the authors are worried about the effect augmentative biological control can have on natural populations of *A. gifuensis* when mass-reared parasitoids are released in the field. However, more antecedents are needed on the species' natural history to understand and support the relevance of this kind of study.

Line 56: Replace capacity with performance.

Lines 58-60: I think the worrying should not be set only on the genetic diversity but on the parasitoid performance too (i.e., infectivity and virulence). It is well known that parasitoid fitness, among others (see below), is important for structuring parasitoid populations, as reported in: Henry, L. M., Roitberg, B. D., & Gillespie, D. R. (2008). Host-range evolution in *Aphidius* parasitoids: Fidelity, virulence and fitness trade-offs on an ancestral host. *Evolution*, 62(3), 689-699.

Lines 331-333: Chemical perception genes appear as good candidates to explain the development and

loss of host fidelity (i.e., detection of chemical cues during the parasitoid development and adult emergence); hence, the authors could analyze diversity depending on the host plant where aphids were sampled.

Lines 339-341: Alternatively, decreased population diversity observed may reflect the action of strong selection on some traits related to host finding or other relevant traits to cope with biotic and abiotic factors. For instance, did the author find some variation in genes coding for detoxification enzymes? It could be interesting to explore detoxifying genes, as the green peach aphid is mainly controlled by frequent applications of synthetic insecticides, so having a look at those genes in parasitoids exposed to sublethal concentrations could give insights into the impact of anthropic effects on population structuring.

Lines 342-344: This has also been recommended for the aphid parasitoid *Aphidius ervi* (see Sepulveda et al., 2017).

Lines 353-354: This seems hard to test without considering the role of multiple biotic (e.g., the aphid host-plant; intraguild predation; the presence of defensive bacterial symbionts in aphids), chemical (e.g., insecticides), or abiotic (e.g., correlational studies against detailed climate data) factors on the genetic features of parasitoid populations. Indeed, there is plenty of evidence that points out the role of those factors in generating local adaptations in parasitoids, particularly in the genus *Aphidius*, as reported in:

Hafer-Hahmann, N., & Vorburger, C. (2020). Parasitoids as drivers of symbiont diversity in an insect host. *Ecology letters*, 23(8), 1232-1241.

Dennis, A. B., Patel, V., Oliver, K. M., & Vorburger, C. (2017). Parasitoid gene expression changes after adaptation to symbiont-protected hosts. *Evolution*, 71(11), 2599-2617.

Hopper, K. R., Oppenheim, S. J., Kuhn, K. L., Lanier, K., Hoelmer, K. A., Heimpel, G. E., ... & Heraty, J. M. (2019). Counties not countries: Variation in host specificity among populations of an aphid parasitoid. *Evolutionary Applications*, 12(4), 815-829.

Zepeda-Paulo, F., Lavandero, B., Mahéo, F., Dion, E., Outreman, Y., Simon, J. C., & Figueroa, C. C. (2015). Does sex-biased dispersal account for the lack of geographic and host-associated differentiation in introduced populations of an aphid parasitoid? *Ecology and Evolution*, 5(11), 2149-2161.

Frago, E. (2016). Interactions between parasitoids and higher order natural enemies: intraguild predation and hyperparasitoids. *Current Opinion in Insect Science*, 14, 81-86.

Desneux, N., Pham-Delègue, M. H., & Kaiser, L. (2004). Effects of sub-lethal and lethal doses of lambda-cyhalothrin on oviposition experience and host-searching behaviour of a parasitic wasp, *Aphidius ervi*. *Pest Management Science: formerly Pesticide Science*, 60(4), 381-389.

Andrade, T. O., Krespi, L., Bonnardot, V., van Baaren, J., & Outreman, Y. (2016). Impact of change in winter strategy of one parasitoid species on the diversity and function of a guild of parasitoids. *Oecologia*, 180, 877-888.

Line 394: As you have obtained parasitoids from sampled aphid mummies, do you have evidence of hyperparasitism of *M. persicae* in China? If so, how did you identify *A. gifuensis* from other emerging parasitoids? When (the season) the parasitoids were collected from natural populations? On what host plants were the aphid mummies sampled? The green peach aphid is well-known due to its wide host range (i.e., it is a high generalist aphid); this is relevant when studying the genetic structure of a parasitoid population due to host fidelity can be developed depending on which host plant the aphid feed and the parasitoid emerge. Indeed, parasitoid wasps (including those from the *Aphidius* group) can learn olfactory and visual cues from the aphid–host–plant interaction during the larval stage and adult emergence. Hence, host fidelity can significantly reduce gene flow between populations because many insects feed, mate, and oviposit on or near their hosts. Therefore, locally adapted gene pools can be degraded when gene flow is high between populations and conserved when gene flow is limited (Henry et al., 2008). It is an aspect not discussed in the manuscript.

Lines 394-399: Did you compute the male/female ratio? I understand only female parasitoids were included in the population genomic survey because females are diploid, and they parasitize aphids. Still, the sex ratio can also give insights into the effects of inbreeding. Indeed, it is known that inbreeding (and other factors like the aphid host size) can alter the sex ratio by increasing the proportion of emerged males, as reported in:

Heimpel, G. E., & Lundgren, J. G. (2000). Sex ratios of commercially reared biological control agents. *Biological Control*, 19(1), 77-93.

Jarošík, V., Holý, I., Lapchin, L., & Havelka, J. (2003). Sex ratio in the aphid parasitoid *Aphidius colemani* (Hymenoptera: Braconidae) in relation to host size. *Bulletin of Entomological Research*, 93(3), 255-258.

Kalule, T., & Wright, D. J. (2005). Effect of cultivars with varying levels of resistance to aphids on development time, sex ratio, size and longevity of the parasitoid *Aphidius colemani*. *BioControl*, 50, 235-246.

Line 408: What taxonomic key was used to determine *Aphidius* species?

Lines 408-409: Add a reference here.

Line 426: How did you manage SNPs from coding and non-coding regions? If the authors are interested in studying the impact of natural selection, it is important to identify neutral variation from putative regions under selection.

Reviewer #2:
Remarks to the Author

TO AUTHORS

GENERAL COMMENTS:

In this manuscript, the authors aim at assessing the impact of biological control populations on wild populations of the parasitoid wasp *Aphidius gifuensis* in China. To do this, they used a population genomics approach, resequencing 542 individuals, including 265 wild individuals, and working with more than 1.8 million SNPs. The manuscript is generally well written and relatively clear in its structure. The question addressed seems to me to be of great interest and the study presented here is promising. However, there are some major flaws that I would like the authors to address thoroughly. Please see below for details.

MAJOR POINTS:

My main concerns with this article are: (i) the lack of precision in certain aspects of the work, (ii) the weaknesses of certain analyses and (iii) the (over-)interpretation of the results.

1. Lack of precision and/or information

A certain amount of important information is missing for a proper understanding of the work carried out. In some cases, this may be known information that the authors simply need to add, but in other cases it seems that the information simply doesn't exist, which may be an issue.

For example, I would like to know more about the history and practice of biological control with *Aphidius gifuensis*. Line 398 mentions "long-term release application of *A. gifuensis*". Can you be more

specific? Overall, I'm not convinced by the description of "post-release populations". What do you mean by "long-term"? An indication of duration (how many years/decades) seems essential to me. And apart from duration, what is the intensity of the releases? How many individuals are typically released per season? Is it a large number of individuals? In general, what are the habits in the fields and are there any differences between the different sites sampled? There are also many unanswered questions about breeding populations. Do all individuals come from the same source population? When was the original sampling done? What are the "15 representative laboratories or factories"? etc. I think all this information is important to put the results obtained into perspective and to better understand the differences between the B, T and N populations. Several sentences in the manuscript suggest that the history of biocontrol is poorly known (e.g. lines 316-318), which further weakens some of the paper's strong conclusions. The authors are kindly requested to provide additional background information and to defend their point of view in a well-argued manner.

Some important elements of the analyses are not sufficiently explained. For example, how are allele frequencies calculated? At the Province level? At the population level? For each of the two wild groups, YUN (e.g. line 519) and non-YUN? Please be as explicit as possible and use the same terms throughout the paper (e.g. "province", "population", etc.). Similarly, in the Δ AFN and Δ AFT calculations, the nature of the allelic frequencies used is not explicit enough. In general, the logic behind these analyses, which include the comparison of spatial expansion processes (Δ AFN) with post-introduction evolutionary processes (Δ AFT), needs to be better explained. Please use clear terms to do this. For example, line 239 "under such circumstances": Which circumstances? Situation 2 or 3, is that it? Not very clear to me.

2. Data analyses

A temporal analysis of the evolution of allele frequencies in the different populations is lacking (e.g. 290-292). Natural selection appears to be able to maintain some of the ancestral polymorphism, but the data presented in this paper do not seem to prove categorically that repeated introductions of artificially bred individuals have no effect on wild populations. An analysis of the genetic diversity of wild populations, particularly in Yunnan, several decades ago would be particularly interesting to assess the long-term effects (e.g. lines 362-363 and lines 373-376). I guess the authors do not have such samples, but I would like them to be less categorical and more cautious in interpreting their results.

Your results show that the Yunnan populations are the oldest, but it is an over-interpretation to say that they represent the ancestral state of the species (e.g. 292-293). I suggest that the authors polarize their alleles using a phylogenetic approach using closely related species (at least *Aphidius ervi*). This will allow them more robust identification of the ancestral alleles and better determine the effect of natural selection on the genetic load, which is the majority of loci under long-term selection within a species. Considering the population of Yunnan as an ancestral state by default seems to me to be a major weakness of the paper.

In the context of the question posed in this article, which focuses in particular on the possible introgression of genes from biological control populations into wild populations, I'm surprised that there aren't certain types of analysis that are quite informative in this context. As I'm not a specialist, I won't go into too much detail, but I'm thinking, for example, of approaches such as those implemented in TREEMIX. The results of such analyses could be informative and of interest to many readers.

Lines 256-257 and lines 535-537: Can you give an in-depth explanation so that everyone can understand the logic behind this proposal? This way of categorizing "strongly" and "weakly affected sites" may seem surprising, especially given that an annotation of SNPs is available. Why don't you use the categorization provided by SNPEFF? Together with the polarization mentioned above, this

would give you access to the evolution of genetic load in a much more efficient and classical way.

3. Interpretation of the results

Overall, I find that the authors repeatedly over-interpret the results of their study. Unless new analyses support their conclusions, please be much less categorical and more cautious in your statements. Below are some examples with comments.

Lines 133-134:

That seems a bit speculative. Wouldn't random gene surfing be enough to explain this pattern? Even if 5 SNPs are involved, is this really an extreme value compared to the size of the gene? Some more information would help to understand this interpretation.

Lines 140-142:

I wonder to what extent an explanation other than adaptation might be at the origin of this pattern. An increase in the nonsyn/syn ratio could also be a sign of an accumulation of genetic load during the spatial expansion of the species - i.e. expansion load. Indeed, the efficiency of purifying selection on invasion fronts is reduced due to recurrent bottlenecks. The results presented here do not seem to me to indicate in any obvious way that this hypothesis should be ruled out. I would like the authors to enlighten me on this particularly important point.

Lines 146-148 and lines 297-298:

Too speculative. Please, be more measured in your conclusions.

Lines 298-300:

In my view, this work provides the beginnings of an answer to the question posed, but does not allow us to draw the categorical conclusions suggested by the authors throughout the manuscript.

Lines 323-325:

I'm not convinced by this conclusion. The causes of the differences observed between wild and bred populations on these 197 candidate genes do not seem clear to me. The natural environment and the laboratory environment are very different by nature, and selection can be applied very differently in the two cases. So is it specific adaptation to the artificial environment? Or the fixation of deleterious alleles linked to the demography of the artificial breeding? Or why not the purging of deleterious alleles in the same breeding context? The data presented here do not provide a clear answer to these questions. The sentence "197 genes would be selected" remains extremely vague, and deserves, at least, to be better justified. You seem to be referring to a fixation of genetic load in breeding populations, but it's never quite made explicit.

MINOR POINTS:

Line 33:

Please change "suppress" to "control".

Lines 34-35:

This sentence is only true in the case of augmentative biocontrol strategies. Please be more specific so as not to mislead people who are not specialists in biological control.

Line 37:

"developed or modified via artificial selection to meet human needs ": I don't think that's the usual way of doing things in the context of biocontrol. If I am wrong, please, provide appropriate references.

Lines 40-42:

Again, you should better explain the context of your study, which focuses on augmentative biological control. The problems outlined here do not apply to classical biological control where there is no wild population at the introduction site. I think it's important to clarify these points as the manuscript is not aimed solely at biological control specialists.

Lines 55-57:

Do you have any references for these statements?

Lines 64-68:

Some of the "four questions" are not presented as questions. Please, try to be consistent.

Lines 74-78:

I'd really like to see a map showing the distribution of the three population types (N, B and T). This map could be included as additional data.

Lines 91-92:

The structure of this sentence suggests that SNPs are only used for evolutionary history inferences. This is (fortunately) not the case: please be clearer about the use of the different types of markers. Simply remove "respectively".

Figure 1:

It is barely indicated in the figure legend that it corresponds to wild individuals only. Please change the title to "Population structure and demographic history of wild *A. gifuensis* populations..."

Figure 1a:

The grey levels on the map seem unnecessary. I suppose it's the altitude, but why would this information be relevant? Please keep the map as simple as possible.

Lines 95-96:

Please change "those from Clusters 1 and 2 were endemic to Yunnan Province " to "those from Clusters 1 and 2 were found only in Yunnan Province".

Lines 106-108:

This statement is not easy to visualise on the ADMIXTURE plot because you use province code in one case and population ID in the other. I suggest that you try to standardise the use of names of geographical locations or populations. At the very least, the population ID could be the same as the province code with an additional number to distinguish them.

In addition, you mention "K=4" at the beginning of the sentence while you refer the reader not only to the ADMIXTURE figure but also to the PCA analysis figure (Fig. 1d), which is not related to a supervised analysis with a predetermined number of clusters. I suggest you remove the "K=4" information to let it only in the figure legend.

Figure 1f:

Please, explain what the grey area is in the figure legend.

Lines 118-120:

Lack of information to understand this statement. I guess the grey area is the LGM. If that is the case, your statement is not obvious to me. Please be more specific.

Line 120:

"(LGM)": I don't think you use this acronym anywhere else in the manuscript. Please remove.

Lines 125-126:

"in each local population by each city": Please be more specific and consistent. This description lacks

clarity insofar as you use the terms "province code" or "population ID" elsewhere.

Figure 2a:

The label "Frequency" should not be use in the Y-axis. I guess this is a number or density of SNPs. Please change accordingly.

Line 139:

"nonsynonymous/synonymous (Nonsyn/Syn) ratios ": can you please describe this statistic in more detail? I can't find a description of it in the methods section. I assume it is the ratio between the number of non-synonymous and synonymous polymorphisms within populations, often referred to as Pn/Ps. If this is the case, please state it clearly.

Supplementary Data 6:

Please, can you indicate the meaning of each column (with units) within the supplementary data?

Lines 158-160 and Figure 2f:

I find this statement lacking in justification. I would appreciate some statistical support for the idea, which is ultimately very visual, that the LG06 chromosome deserves to be looked at more closely.

Lines 171-172:

"shared similar genetic structure" with what? Please, be more specific.

Line 182:

"(pairwise-FST = 0.0121)": Is this a mean pairwise-FST. Please be more specific.

Line 197:

Please change "perfomed" to "performed".

Lines 197-198:

Do you have a reference for this statement?

Line 249:

"More profound" than what? What impacts? Please be more specific.

Figure 4e:

The nonsyn/syn values are lower for "weakly affected sites" than for "strongly affected sites". Can you explain this?

Line 308:

"the genetic drift of inbreeding": this is a peculiar expression. I guess "genetic drift" would be sufficient here.

Lines 308-309:

"individuals within populations were genetically more similar to one another than individuals from external populations": A very circular phrase, in that the notion of genetic proximity between individuals is an integral part of the very definition of what a 'population' is. The whole sentence is of little interest, as it merely states the obvious.

Line 309:

Please change "t" to "to".

Lines 313-315:

Unclear. Please rephrase.

Lines 319-321:

Again, I would like a reference for this statement.

Line 340:

"selected genes": This is not clear. You refer to "genes under selection", is that it? Anyway, what you describe as genetic erosion is mainly driven by genetic drift, not selection, so you should be more careful with the terms you use.

Lines 342-344:

This may seem a reasonable proposition, but how does it relate to the results presented in this manuscript? Firstly, the impact of biological control on the wild gene pool is considered to be of little importance here. Second, none of the results presented here indicate whether the genetic composition of breed individuals is detrimental in the very peculiar context of breeding environments and human agricultural activity (i.e. rapid and effective control of pest populations does not necessarily require the same traits as in the wild).

You refer to the risks for survival performance, but is there any signs of any problems with the breeding of this species?

Line 355:

"severely affected": I'm not a big fan of this term, which seems to me to refer mainly to a decline in diversity. Please be more precise in the terms you use.

Lines 388-390:

On what basis do you consider it to be a good model? It's certainly an interesting case study, but why "model"? I suppose you could consider the species as a model if you had a large ancestral sample, significant genomic resources, and the species was particularly easy to sample. As all these conditions are not met, I would avoid using this term, which unnecessarily overstates the results of the study.

Lines 393-404:

Thank you for clarifying here that these are only females (see line 74). This is important because it has a strong impact on sample size, as it seems to me that the species is haplo-diploid.

Line 396:

What are the criteria for defining "areas without release records"? In particular, is there a minimum distance between a known release site and such areas for them to be considered as such? If such a criterion exists, is it based on the known dispersal characteristics/abilities of the species?

Lines 402-403:

Why do you say "additional samples"? As I understand it, these 162 individuals are part of the populations B, T and N presented above. The whole sentence can be deleted to avoid confusion. Or it could simply be pointed out that Yunnan is over-represented in the sample for historical reasons. Incidentally, these historical reasons are not clear to me: is the species more abundant in this province?

Line 450:

Please remove "Thus".

Line 457:

Please write "4.FST"

Lines 462-463:

How were the K shown in Figure 1c selected? If the choice was made by the authors on the basis of visual criteria, it is important to present the figures with all values of K as additional information.

Line 477:

"MCMC": don't you mean MSMC?

Line 478:

Please change "in the bumblebee" to "as estimated for the bumblebee".

Line 479:

Do you have any reference for this? 15 generations seems like a lot.

Lines 511-512:

"we selected 35 wild populations (N)": As I understand it, the whole wild sample is made up of 35 populations (supplementary data 1), isn't that right? Shouldn't it be "we used the 35 wild populations (N)"?

Lines 523-524:

"AFA < 0.01": According to Supplementary Data 1, the maximum number of individuals is 14, which gives a minimum possible frequency of 1/28, in other words 0.036. This threshold of 0.01 corresponds to populations with an estimated allelic frequency of 0, doesn't it? If so, this criteria is somewhat misleading. Why not simply say that you are considering SNPs whose frequencies are equal to 0 in the YUN populations? However, it is possible that the AFs are calculated on all pooled samples. If this is the case, you need to be much clearer, particularly in the use of terms, for example by a better use of the terms "province" and "population", as in Supplementary Data 1 and 2.

Lines 526-527:

Again, so the AF is measured at the Province level? Or at the population level? This is unclear in the whole AF description section. Please be more specific.

Line 528:

Please change "number of sites" to "number of SNP sites".

Line 538:

I'm not sure I understand what you mean by "simulation". Can you elaborate?

Line 544:

"in comparison to *Aphidius ervi*": I assume you're referring to the *Aphidius ervi* genome. Please be more specific and indicate which version of the genome you are using.

RESPONSE TO REVIEWER COMMENTS

Reviewer #1

In this manuscript, Bingyan Li et al. report a population genomic survey of the aphid parasitoid *Aphidius gifuensis* in China to determine whether the release of mass-reared parasitoid wasps may alter the genetic features of natural populations of the same species. The authors are particularly interested in studying the effects of inbreeding on parasitoid performance at the field scale, and they looked for genomic evidence to support that idea. The work included the sampling of wild and cage-reared populations, also including samples of parasitoids from an aphid population that is currently under biological control. The authors generated a massive amount of data from genome resequencing. Then they described the data with no clear hypothesis other than stating that natural selection maintains genetic diversity in wild populations. However, they don't discuss what selection agents are putatively acting on populations and their significance. Hence, my main concern is related to data interpretation and discussion, as in its present status, the manuscript is too descriptive, not going into details about how the biology of the parasitoid and the role of its interactions with other organisms and the environment can explain the genetic diversity, as I review below. Consequently, based on what I read, I do not recommend this manuscript to be accepted in its present form, as much work is needed to be of wide interest to the readers of Nature Communications.

RESPONSE: Thank you very much for your suggestions and comments about our manuscript. We have extensively revised the manuscript according to these suggestions and described our responses in this letter. Generally, in the new version, we mentioned that both the biotic (host plant) and abiotic (climate) factors were the major selection agents (Lines 283-291). Moreover, the strength of conclusion was weakened in some parts with insufficient evidence (Lines 524-529).

Lines 34-35: The statement is not entirely accurate as there are examples of successful biological control programs of aphids based on single introduction events with no further releases of biocontrol agents (e.g., parasitoid wasps to control cereal aphids in South America), as reviewed in:

Hance, T., Kohandani-Tafresh, F., and Munaut, F. (2017). Biological control. In *Aphids as crop pests*, H. Van Emden and R. Harrington, eds (CABI Publishing), pp. 448–493.

RESPONSE: We are grateful for the correction. We have revised this sentence and pointed out that the strategy of mass rearing and the periodic release is applicable to augmentative biocontrol. (Lines 34-38)

Lines 37-40: There are examples of aphid parasitoids losing specialization due to inbreeding. For instance, host fidelity (i.e., parasitoids prefer the natal aphid–host–plant system from where they emerged and increase their foraging efficiency) can be lost due to inbreeding, as reported

in:

Sepúlveda, D. A., Zepeda-Paulo, F., Ramírez, C. C., Lavandero, B., & Figueroa, C. C. (2017). Loss of host fidelity in highly inbred populations of the parasitoid wasp *Aphidius ervi* (Hymenoptera: Braconidae). *Journal of Pest Science*, 90, 649-658.

RESPONSE: We added the description about the influence of inbreeding on host fidelity of parasitoid wasps in detail. (Lines 43-44)

Line 44: You mean balancing selection? What selection agents may be involved? Explain a bit more about how and under what circumstances natural selection can preserve the genetic diversity of a population.

RESPONSE: We added more explanations about natural selection preserving population genetic diversity. There are examples of balancing selection that play a role in maintaining the genetic diversity and gene pool of wild insect populations (Koenig et al., 2019; Kumar et al., 2019). (Lines 55-63)

Line 47: Replace abnormal with increased.

RESPONSE: Corrected accordingly. (Line 64)

Lines 51-52: Where is this parasitoid species originated? I assume the species is endemic to East Asia, and that's why the authors are worried about the effect augmentative biological control can have on natural populations of *A. gifuensis* when mass-reared parasitoids are released in the field. However, more antecedents are needed on the species' natural history to understand and support the relevance of this kind of study.

RESPONSE: Yes, *A. gifuensis* is endemic to East Asia. It was introduced locally, reared indoors and released in the fields for biocontrol application in China. To better describe reasons for our study, we added the original information of *A. gifuensis* as following “distributes widely in Asia, such as China, Japan and Korea (Umer et al., 2021; Song Yang et al., 2011; Zhang et al., 2018)” and described the augmentative biocontrol of *A. gifuensis* with its potential effects in detail. (Lines 68-83)

Line 56: Replace capacity with performance.

RESPONSE: We have corrected it accordingly. (Line 80)

Lines 58-60: I think the worrying should not be set only on the genetic diversity but on the

parasitoid performance too (i.e., infectivity and virulence). It is well known that parasitoid fitness, among others (see below), is important for structuring parasitoid populations, as reported in:

Henry, L. M., Roitberg, B. D., & Gillespie, D. R. (2008). Host-range evolution in *Aphidius* parasitoids: Fidelity, virulence and fitness trade-offs on an ancestral host. *Evolution*, 62(3), 689-699.

RESPONSE: Thanks for your suggestion. It is evident that long-term breeding can influence various traits in parasitoid wasps, like parasitoid performance, host fidelity or environmental adaptation. Our study did not delve extensively into these effects, representing a limitation. The primary objective of our study was to investigate the genetic consequences of prolonged captive breeding and continuous release on wild populations, guided by prior research highlighting the phenomenon of population degradation and diminished parasitoid efficacy in artificially bred populations. In our future studies, we would explore the specific impacts of long-term rearing on the parasitoid performance of *A. gifuensis* in detail, addressing these important factors comprehensively and scientifically.

Lines 331-333: Chemical perception genes appear as good candidates to explain the development and loss of host fidelity (i.e., detection of chemical cues during the parasitoid development and adult emergence); hence, the authors could analyze diversity depending on the host plant where aphids were sampled.

RESPONSE:

Thanks for your suggestion. It is expected that the nucleotide diversity should be higher for the wild population compared with the bred or released population. Therefore, to identify the potential relationship between chemical perception genes and the host fidelity, we carried out GWAS among the individuals collected from different host plants (see Methods for details). Pairwise comparisons were performed between four host plants *Nicotiana*, *Brassica*, *Capsicum*, and *Solanum*. We selected the top 5% of the most significant sites of each comparison, but no chemical perception genes were found among the host genes of the SNPs (Fig. S11 and Supplementary Data 13). This suggests that in our cases, chemical perception genes might not be the most essential (top 5%) determinant of host fidelity, or some unknown technical issues limited the detection power. However, we still reserve the possibility that the chemical perception genes might partially contribute to the adaptation to different host plants. We added the above contents to the according part in the Discussion section. (Lines 463-473)

Lines 339-341: Alternatively, decreased population diversity observed may reflect the action of strong selection on some traits related to host finding or other relevant traits to cope with biotic and abiotic factors. For instance, did the author find some variation in genes coding for detoxification enzymes? It could be interesting to explore detoxifying genes, as the green peach aphid is mainly controlled by frequent applications of synthetic insecticides, so having a look at those genes in parasitoids exposed to sublethal concentrations could give insights into the

impact of anthropic effects on population structuring.

RESPONSE: This is a good point. We followed this idea and identified several genes playing important roles in regulating growth and reproduction of *A. gifuensis* based on selective sweep analysis. These genes included 20 genes participated in response to external signals, 8 chemical perception genes, 3 enriched in cyclin-dependent protein kinase activity and 8 lifespan-associated genes (Supplementary Data 9-11). These genes reflected the selection on some traits related to host finding or other relevant traits to cope with biotic and abiotic factors. We also identified a detoxifying gene, ABC transporter G family (ABCG), which displayed differences between wild and artificially bred populations. Additionally, we included a discussion regarding the ABCG gene and its potential role in insecticide resistance. However, the artificially bred populations are not exposed with chemical pesticides. And due to the lack of chemical control records in the open fields, we are unable to determine whether the varying diversity of ABCG gene between *A. gifuensis* wild and artificially bred populations was associated with insecticide resistance at this stage. (Lines 473-480)

Lines 342-344: This has also been recommended for the aphid parasitoid *Aphidius ervi* (see Sepulveda et al., 2017).

RESPONSE: We added the reference about the example of the aphid parasitoid *Aphidius ervi*. (Lines 488-491)

Lines 353-354: This seems hard to test without considering the role of multiple biotic (e.g., the aphid host-plant; intraguild predation; the presence of defensive bacterial symbionts in aphids), chemical (e.g., insecticides), or abiotic (e.g., correlational studies against detailed climate data) factors on the genetic features of parasitoid populations. Indeed, there is plenty of evidence that points out the role of those factors in generating local adaptations in parasitoids, particularly in the genus *Aphidius*, as reported in:

Hafer-Hahmann, N., & Vorburger, C. (2020). Parasitoids as drivers of symbiont diversity in an insect host. *Ecology letters*, 23(8), 1232-1241.

Dennis, A. B., Patel, V., Oliver, K. M., & Vorburger, C. (2017). Parasitoid gene expression changes after adaptation to symbiont-protected hosts. *Evolution*, 71(11), 2599-2617.

Hopper, K. R., Oppenheim, S. J., Kuhn, K. L., Lanier, K., Hoelmer, K. A., Heimpel, G. E., ... & Heraty, J. M. (2019). Counties not countries: Variation in host specificity among populations of an aphid parasitoid. *Evolutionary Applications*, 12(4), 815-829.

Zepeda-Paulo, F., Lavandero, B., Mahéo, F., Dion, E., Outreman, Y., Simon, J. C., & Figueroa, C. C. (2015). Does sex-biased dispersal account for the lack of geographic and host-associated differentiation in introduced populations of an aphid parasitoid? *Ecology and Evolution*, 5(11), 2149-2161.

Frago, E. (2016). Interactions between parasitoids and higher order natural enemies: intraguild predation and hyperparasitoids. *Current Opinion in Insect Science*, 14, 81-86.

Desneux, N., Pham-Delègue, M. H., & Kaiser, L. (2004). Effects of sub-lethal and lethal doses

of lambda-cyhalothrin on oviposition experience and host-searching behaviour of a parasitic wasp, *Aphidius ervi*. *Pest Management Science: formerly Pesticide Science*, 60(4), 381-389.

Andrade, T. O., Krespi, L., Bonnardot, V., van Baaren, J., & Outreman, Y. (2016). Impact of change in winter strategy of one parasitoid species on the diversity and function of a guild of parasitoids. *Oecologia*, 180, 877-888.

RESPONSE: Thanks for raising this good suggestion and providing those literatures. We have revised the conclusion and added more detailed discussion to the manuscript. We collected female adults from the same host aphid, *M. persicae*, and conducted the genetic comparisons between populations from the same region or with the same population structure, to avoid the influence of host species and geographic location. And we intend to further investigate the factors, traits and their associated genes that could contribute to differences in population genetic variations of *A. gifuensis*. (Lines 501-508)

Line 394: As you have obtained parasitoids from sampled aphid mummies, do you have evidence of hyperparasitism of *M. persicae* in China? If so, how did you identify *A. gifuensis* from other emerging parasitoids? When (the season) the parasitoids were collected from natural populations? On what host plants were the aphid mummies sampled? The green peach aphid is well-known due to its wide host range (i.e., it is a high generalist aphid); this is relevant when studying the genetic structure of a parasitoid population due to host fidelity can be developed depending on which host plant the aphid feed and the parasitoid emerge. Indeed, parasitoid wasps (including those from the *Aphidius* group) can learn olfactory and visual cues from the aphid–host–plant interaction during the larval stage and adult emergence. Hence, host fidelity can significantly reduce gene flow between populations because many insects feed, mate, and oviposit on or near their hosts. Therefore, locally adapted gene pools can be degraded when gene flow is high between populations and conserved when gene flow is limited (Henry et al., 2008). It is an aspect not discussed in the manuscript.

RESPONSE: Thanks for raising these questions. The main hyperparasitism of *M. persicae* is *Pachyneuron aphidis* (Bouche). We identified *A. gifuensis* from other parasitoid wasps through morphology and DNA barcoding as described in the Methods section (Lines 584-587). The detail information about the collection time (almost between May and July) of wild populations and the host plant species were listed in Supplementary Data 1 and 2. Before collecting samples, we considered the influences of geographical distribution and various host plants on population genetic structure of *A. gifuensis*. Thus, we collected wild populations from different host plants, including *Brassica chinensis*, *Brassica napus*, *Brassica pekinensis*, *Capsicum annuum*, *Lycopersicon esculentum*, *Nicotiana tabacum*, *Solanum melongena* and *Solanum tuberosum*. All artificially bred and post-release individuals were collected from tobacco fields. However, based on the results of wild population structure, populations collected from different host plants exhibited similar structure except YUN populations (Fig. 1c), reflecting high gene flow between the peripheral populations (non-YUN). Host plant species was not the main factor contributing the population structure of *A. gifuensis*.

Lines 394-399: Did you compute the male/female ratio? I understand only female parasitoids were included in the population genomic survey because females are diploid, and they parasitize aphids. Still, the sex ratio can also give insights into the effects of inbreeding. Indeed, it is known that inbreeding (and other factors like the aphid host size) can alter the sex ratio by increasing the proportion of emerged males, as reported in:

Heimpel, G. E., & Lundgren, J. G. (2000). Sex ratios of commercially reared biological control agents. *Biological Control*, 19(1), 77-93.

Jarošík, V., Holý, I., Lapchin, L., & Havelka, J. (2003). Sex ratio in the aphid parasitoid *Aphidius colemani* (Hymenoptera: Braconidae) in relation to host size. *Bulletin of Entomological Research*, 93(3), 255-258.

Kalule, T., & Wright, D. J. (2005). Effect of cultivars with varying levels of resistance to aphids on development time, sex ratio, size and longevity of the parasitoid *Aphidius colemani*. *BioControl*, 50, 235-246.

RESPONSE: Thanks for this comment. Generations of inbreeding can evidently alter the sex ratios of progeny of parasitoid wasps. Although samples (mummied aphids) were collected randomly in the fields, the male/female ratio of these samples cannot represent the truth sex ratio in wild or artificially bred environment. We did not record this information and therefore are unable to use sex ratio to infer the effect of inbreeding. On the other hand, our study originally aimed to investigate the genetic influences on *A. gifuensis* individuals that have undergone mass rearing for over 20 years in China. Thus, we chose diploid female individuals to obtain a more comprehensive set of genetic information.

Line 408: What taxonomic key was used to determine *Aphidius* species?

RESPONSE: We added the reference of the taxonomic key used to identified sample as a member of *Aphidius*. (Lines 585). The samples were finally identified into species level based on DNA barcoding sequences (partial mitochondrial *cox1* gene).

Lines 408-409: Add a reference here.

RESPONSE: Added accordingly. (Line 587)

Line 426: How did you manage SNPs from coding and non-coding regions? If the authors are interested in studying the impact of natural selection, it is important to identify neutral variation from putative regions under selection.

RESPONSE: For selective sweep analysis, we firstly identified the regions under selection using three different methods, including genome-wide distribution of population fixation statistics F_{ST} , nucleotide diversity $\theta\pi$ and the XP-CLR approach. Meanwhile, another

independent tool SNPEff was used to annotate each SNP. SNPEff tells us the location of each SNP (intergenic, exonic, CDS, or UTR), and if in CDS, is it a missense (Nonsyn) or synonymous site, leading to what amino acid changes. Notably, selected region and coding region are different concepts defined by different approaches. For the suggestion raised by the reviewer, we have used neutral (synonymous) variations as control to measure the natural selection on nonsynonymous sites (Fig 2d).

Again, we thank you for your time and effort in reviewing this manuscript.

RESPONSE TO REVIEWER COMMENTS

Reviewer #2

GENERAL COMMENTS:

In this manuscript, the authors aim at assessing the impact of biological control populations on wild populations of the parasitoid wasp *Aphidius gifuensis* in China. To do this, they used a population genomics approach, resequencing 542 individuals, including 265 wild individuals, and working with more than 1.8 million SNPs. The manuscript is generally well written and relatively clear in its structure. The question addressed seems to me to be of great interest and the study presented here is promising. However, there are some major flaws that I would like the authors to address thoroughly. Please see below for details.

RESPONSE: We sincerely appreciate the kind review and insightful comments by the reviewer. We have revised our manuscript considering all the points. Please see our detailed response below.

MAJOR POINTS:

My main concerns with this article are: (i) the lack of precision in certain aspects of the work,
(ii) the weaknesses of certain analyses and (iii) the (over-)interpretation of the results.

1. Lack of precision and/or information

A certain amount of important information is missing for a proper understanding of the work carried out. In some cases, this may be known information that the authors simply need to add, but in other cases it seems that the information simply doesn't exist, which may be an issue.

RESPONSE: We are grateful for these suggestions. We have added some detailed information to the main text or supplementary material. For some missing information, we adjusted the according conclusion to be less categorical. The detailed responses are as follows.

For example, I would like to know more about the history and practice of biological control with *Aphidius gifuensis*.

RESPONSE: More detailed information about the history and practice of biological control with *Aphidius gifuensis* were described in the Introduction section. In China, this parasitoid wasp has been utilized to control green peach aphid populations in Yunnan Province since 1997. Thus, Yunnan province served as the primary region for our research, where we conducted analyses to assess the genetic influences of long-

term mass rearing. Additionally, it's worth noting that this parasitoid wasp undergoes 15-20 generations per year, resulting in hundreds of generations over the course of more than 20 years. (Lines 68-83)

Line 398 mentions "long-term release application of *A. gifuensis*". Can you be more specific? Overall, I'm not convinced by the description of "post-release populations". What do you mean by "long-term"? An indication of duration (how many years/decades) seems essential to me. And apart from duration, what is the intensity of the releases? How many individuals are typically released per season? Is it a large number of individuals? In general, what are the habits in the fields and are there any differences between the different sites sampled?

RESPONSE: We have included the time when *A. gifuensis* was introduced for application in Supplementary Data 2, which spans from approximately 1997 to 2017. Besides, it is consistently released into open fields during seasons with high aphid infestations each year. Two or three releases of *A. gifuensis* were conducted at different growth stages of tobacco, involving 10,000-15,000 individuals per hectare for each release. Thus, the post-release populations were collected after the release application during the period between June and August in tobacco fields. These populations may consist of a mixture and hybrid of bred and wild individuals. *A. gifuensis* typically have around 15 generations per year. Natural selection and the shift in allele frequency could occur within even 2~3 years. This period could be regarded as long-term for *A. gifuensis*. Therefore, our data are suitable for investigating the effect of released population on the gene pool of wild populations. (Lines 569-572)

There are also many unanswered questions about breeding populations. Do all individuals come from the same source population? When was the original sampling done? What are the "15 representative laboratories or factories"? etc.

I think all this information is important to put the results obtained into perspective and to better understand the differences between the B, T and N populations.

RESPONSE: The introduced origin of each artificially bred population was listed in Supplementary Data 2. The artificially bred populations of Guangdong, Hunan and Sichuan originated from a common source population located in Yuxi (Yunnan). However, in most cases, populations were introduced locally.

The "15 representative laboratories or factories" indicated that the sample collection of artificially bred populations almost covered the laboratories or factories of mass rearing in China. The original expression may not be suitable and we have deleted "representative".

Several sentences in the manuscript suggest that the history of biocontrol is poorly

known (e.g. lines 316-318), which further weakens some of the paper's strong conclusions. The authors are kindly requested to provide additional background information and to defend their point of view in a well-argued manner.

RESPONSE: Thanks for raising this concern. We have revised these sentences to discuss how the differential origins of the introduced population may affect the genetic diversity and inbreeding level among artificially bred populations. (Lines 437-441) On the other hand, our main conclusion, that natural selection preserved the gene pool of wild population, was supported by the data in two independent provinces with sufficient individuals (HEB and HEN, the old Figure 4). In the revised manuscript, we tried our best to provide additional background information as we responded to the previous concerns.

Some important elements of the analyses are not sufficiently explained. For example, how are allele frequencies calculated? At the Province level? At the population level? For each of the two wild groups, YUN (e.g. line 519) and non-YUN? Please be as explicit as possible and use the same terms throughout the paper (e.g. "province", "population", etc.).

RESPONSE: We apologize for the insufficient explanation. The AF was calculated for each province. In the revised manuscript, we have used province instead of population when necessary. Particularly, in the beginning of the “*Natural selection restricts the post-release effects on wild populations*” section, we stressed that “The following interpretation and analyses are described at province level”. (Lines 301-302)

Similarly, in the ΔAF_N and ΔAF_T calculations, the nature of the allelic frequencies used is not explicit enough. In general, the logic behind these analyses, which include the comparison of spatial expansion processes (ΔAF_N) with post-introduction evolutionary processes (ΔAF_T), needs to be better explained. Please use clear terms to do this. For example, line 239 “under such circumstances”: Which circumstances? Situation 2 or 3, is that it? Not very clear to me.

RESPONSE: We apologize for the ambiguity. In the revised manuscript, to make our analysis more understandable and more quantitative, we slightly amended our methodology on the comparison between ΔAF of different sites. Meanwhile, we also added schematic diagrams to illustrate our logic (new Fig. 5a-5c). This reviewer understood ΔAF_N as the changes during spatial expansion processes. Actually, we aimed to use ΔAF_N to represent the “temporal” evolutionary processes. We apologize for the confusion. The rationale was explained in new Fig. 5a and the corresponding text. Notably, as the reviewer requested, we used YUN population plus the outgroup

species (*A. ervi*) to infer the ancestral state as well as the direction of mutations (new Fig. 5a). E.g. $AF_A = 0$ means that all the alleles in YUN wild individuals and the *A. ervi* sequence were identical to the reference genome of *A. gifuensis* (new Fig. 5a).

2. Data analyses

A temporal analysis of the evolution of allele frequencies in the different populations is lacking (e.g. 290-292). Natural selection appears to be able to maintain some of the ancestral polymorphism, but the data presented in this paper do not seem to prove categorically that repeated introductions of artificially bred individuals have no effect on wild populations. An analysis of the genetic diversity of wild populations, particularly in Yunnan, several decades ago would be particularly interesting to assess the long-term effects (e.g. lines 362-363 and lines 373-376). I guess the authors do not have such samples, but I would like them to be less categorical and more cautious in interpreting their results.

RESPONSE: Indeed, we lacked historic records and temporal data. We have discussed that the absence of historic data (e.g. the genetic information of a population collected several decades ago) may hinder the temporal analysis of allele frequency dynamics. Therefore, we rely on YUN plus outgroup *Aphidius ervi* information to represent the ancestral state of non-YUN populations.

In the revised manuscript, we tried to be less categorical and more cautious in interpreting our results. “However, the lack of historical data, such as the genomic information of wild populations prior to the application of biocontrol agents, hampered the quantitative assessment of the evolutionary consequences of release. Thus, our analyses were not the direct inspection of the temporal evolutionary process, and there was the possibility that some alternative explanations accounting for our observations. Deeper works in the future are needed to give a more solid and confirmative conclusion.”. (Lines 524-529)

Your results show that the Yunnan populations are the oldest, but it is an over-interpretation to say that they represent the ancestral state of the species (e.g. 292-293). I suggest that the authors polarize their alleles using a phylogenetic approach using closely related species (at least *Aphidius ervi*). This will allow them more robust identification of the ancestral alleles and better determine the effect of natural selection on the genetic load, which is the majority of loci under long-term selection within a species. Considering the population of Yunnan as an ancestral state by default seems to me to be a major weakness of the paper.

RESPONSE: Thanks for this suggestion. On one hand, we have reconstructed phylogenetic tree based on nucleic SNPs using *A. ervi* as outgroup and the result was consistent with mitochondrial genomes. The YUN populations could be divided into

two groups and were more ancient than other populations (Fig. S3). On the other hand, in the SNP analyses, we have polarized the direction of the mutation and adjusted the alternative allele frequency to derived allele frequency.

In the context of the question posed in this article, which focuses in particular on the possible introgression of genes from biological control populations into wild populations, I'm surprised that there aren't certain types of analysis that are quite informative in this context. As I'm not a specialist, I won't go into too much detail, but I'm thinking, for example, of approaches such as those implemented in TREEMIX. The results of such analyses could be informative and of interest to many readers.

RESPONSE:

We performed an ABBA-BABA analysis implemented in Dsuite to test the genetic introgression between artificially bred or post-release populations and wild populations using *A. ervi* as an outgroup. Based on synonymous SNPs, significant genetic introgression between artificially bred or post-release populations and wild populations was found in Hebei (HEB), Henan (HEN) and Yunnan (Chuxiong, YUN01) (P -value < 0.05). By contrast, no significant genetic introgression was found based on missense or all SNPs, except Yunnan (Chuxiong, YUN01) (Supplementary Data 12). As missense sites are functionally more essential than synonymous sites, these results suggest that natural selection might have partially suppressed the impact of released population on wild populations. (Lines 392-300)

Lines 256-257 and lines 535-537: Can you give an in-depth explanation so that everyone can understand the logic behind this proposal? This way of categorizing "strongly" and "weakly affected sites" may seem surprising, especially given that an annotation of SNPs is available.

Why don't you use the categorization provided by SNPEFF? Together with the polarization mentioned above, this would give you access to the evolution of genetic load in a much more efficient and classical way.

RESPONSE: Thanks for your question and sorry for the unclear definition. In the revised version, we added several diagrams to explain our definition. Following the definition that $\Delta AF_N = AF_N - AF_A$ in a particular province (Fig. 5a), we try to measure the extent of which the AF_N has been affected by the released population, we first defined SNPs with $\Delta AF_N \geq 0.05$ as affected sites. Then we divided affected sites into "strongly affected sites" and "weakly affected sites" (Fig. 5b). Briefly, SNPs with significantly different AF between AF_N and AF_A were strongly affected sites and the

remaining insignificant SNPs were weakly affected sites (Fig. 5b, see Methods for detail). Intuitively, under random genetic drift (null hypothesis), strongly affected sites should show similar components (e.g., the Nonsyn/Syn ratio) with weakly affected sites (Fig. 5c). However, if natural selection tries to maintain the gene pool in wild population (AF_N), then we expect that the strongly affected sites should show lower fraction of Nonsyn sites compared to the weakly affected sites due to the fact that some Nonsyn mutations should be eliminated by purifying selection before they could reach a remarkable change in allele frequency (Fig. 5c). Under this selection pressure, the gene pool in wild population would be saved. (Lines 333-345).

3. Interpretation of the results

Overall, I find that the authors repeatedly over-interpret the results of their study. Unless new

analyses support their conclusions, please be much less categorical and more cautious in your

statements. Below are some examples with comments.

RESPONSE: Thanks for your comments. We fully realized that we should be more cautious and less categorical in some statement. Here are some examples that we adjusted (weakened) the strength of our conclusions according to your suggestion (Lines 524-529).

Lines 133-134:

That seems a bit speculative. Wouldn't random gene surfing be enough to explain this pattern? Even if 5 SNPs are involved, is this really an extreme value compared to the size of the gene? Some more information would help to understand this interpretation.

RESPONSE: Thanks for your raising this question. Although one gene might have numerous Nonsyn SNPs, not all of them have AF significantly correlated with the longitude or latitude across different provinces. One gene with five such “significant SNPs (Nonsyn)” is very rare. To show how unexpected it is to obtain five (significant) nonsynonymous SNPs in a gene, we first displayed the distribution of CDS lengths of all genes with significant nonsynonymous SNP (Fig. 2c). The CDS length of gene *evm.model.Contig2.505* was almost at the median level (Fig. 2c, 1431 bp, 43.2% quantile). Then we chose the genes with similar CDS length (1000~2000) and calculated how many nonsynonymous SNPs a gene had. We found that in most cases one gene only had one such SNP (Fig. 2c). This suggests that five such SNPs in *evm.model.Contig2.505* should be an extreme value compared with the size of the gene. (Lines 167-173)

Lines 140-142:

I wonder to what extent an explanation other than adaptation might be at the origin of this pattern. An increase in the nonsyn/syn ratio could also be a sign of an accumulation of genetic load during the spatial expansion of the species - i.e. expansion load. Indeed, the efficiency of purifying selection on invasion fronts is reduced due to recurrent bottlenecks. The results presented here do not seem to me to indicate in any obvious way that this hypothesis should be ruled out. I would like the authors to enlighten me on this particularly important point.

RESPONSE: Thanks for your comments. First, we have polarized the direction of the mutation according to your suggestion, that was, when the reference allele was the derived allele and the alternative allele was identical to the sequence in outgroup *A. ervi*, we changed the AF to 1-AF for all populations (provinces). But this adjustment would only change *Rho* to *-Rho* and would not affect the *P* value (significance) in the correlation tests. Therefore, in Fig. 2, only the histogram in Fig. 2a has changed and the other results were not affected.

Notably, the “insignificant” grey bars in original Fig. 2c has been updated due to a flaw in the previous version. We apologize for that. In the updated results, the Nonsyn/Syn ratio for the insignificant group was the lowest, which supported our notion.

Next, we need to answer the reviewer’s question that how to exclude the possibility of genetic load in shaping the pattern in original Fig. 2c (new Fig. 2d). For all pooled wild populations, we calculated the LD (r^2) between a Nonsyn SNP and its nearest Syn SNP (Fig. 2e). Our null hypothesis was, if some observed Nonsyn mutations were resulted from the accumulation of genetic load rather than adaptation, then the LD (r^2) between the Nonsyn site and another neutral (Syn) sites should be random and low (Fig. 2f), and should not show differences across different groups of mutations. However, if part of the Nonsyn SNPs were positively selected during migration, then the LD around the selected Nonsyn sites should be stronger than other non-selected Nonsyn sites (Fig. 2f). Indeed, when we divided the Nonsyn SNPs into those categories defined in Fig. 2c, we saw that the LD (r^2) was actually increasing with $|Rho|$ values (Fig. 2e). This observation rejected the null hypothesis of random accumulation of genetic load. Instead, a more plausible explanation was, the excessive Nonsyn mutations in the “ $|Rho|>0.7$ ” groups were accumulated due to positive selection. Upon rapid positive selection (e.g. during environmental adaptation), the neutral sites (Syn SNPs) around the positively selected Nonsyn SNPs might undergo hitchhiking and show strong linkage with the Nonsyn SNPs. For the less functional or unselected Nonsyn SNPs, their surrounding neutral sites might not show strong linkage with them. Importantly, one may argue that bottleneck effect will also cause strong LD. However, under bottleneck, the strong LD should appear as genomic blocks or even be strong across the whole genome (Fig. 2f), and this is completely different from our observation that the regions of strong LD were scattered and dependent on the correlation with geographical parameters (Fig. 2d and 2e).

Taken together, these results further support our assumption that the Nonsyn SNPs significantly correlated with longitude/latitude (with high $|Rho|$) were likely to be functional and participating in the environmental adaptation during spatial expansion.

Meanwhile, we stress that these conclusions are totally based on *in silico* data and more delicate measurements should be designed to robustly confirm our assumptions. (Lines 180-201)

Lines 146-148 and lines 297-298:

Too speculative. Please, be more measured in your conclusions.

RESPONSE: Thanks for your reminder. The original sentence of lines 146-148, together with the figure it referred to (old Fig. 2d), was removed in the new version. Since we have the detailed functional enrichment of genes of interest (new Fig. 3), we removed the old Fig. 2d which tried to measure the essentiality of genes with their dN/dS values (but the connection of which is indirect).

We changed the sentence of lines 297-298 to “we discovered that excessive nonsynonymous mutations (compared to synonymous mutations) were associated with local geographic features and located in evolutionarily conserved genes, indicating that these nonsynonymous mutations were likely involved in the environmental adaptation during the dispersal of wild *A. gifuensis* population.” (Lines 403-408)

Lines 298-300:

In my view, this work provides the beginnings of an answer to the question posed, but does not allow us to draw the categorical conclusions suggested by the authors throughout the manuscript.

RESPONSE: We have revised this sentence to make it less categorical. The new sentence is as follows: “The population genetic system of wild *A. gifuensis* helps us understand the genetic diversity pattern during rearing and post-release, and might provide an opportunity for measuring the effect of long-term mass rearing and releases on wild populations.” Accordingly, when necessary, we revised the strength of our conclusions throughout the manuscript. (Lines 414-416)

Lines 323-325:

I'm not convinced by this conclusion. The causes of the differences observed between wild and bred populations on these 197 candidate genes do not seem clear to me. The natural environment and the laboratory environment are very different by nature, and selection can be applied very differently in the two cases. So is it specific adaptation to the artificial environment? Or the fixation of deleterious alleles linked to the demography of the artificial breeding? Or why not the purging of deleterious alleles in the same breeding context? The data presented here do not provide a clear answer to these questions. The sentence "197 genes would be selected" remains extremely vague, and deserves, at least, to be better justified. You seem to be referring to a fixation of genetic load in breeding populations, but it's never quite made explicit.

RESPONSE: Thanks for your suggestions. We revised this sentence to make it clearer. The 197 candidate genes were identified to explain the genetic changes associated with mass rearing based on selective sweep analyses. We no longer use the term “selected genes”. Due to the lack of corresponding phenotype traits and the differences between wild and captive environment, it is difficult to distinguish between the multiple possibilities (adaptation to artificial environment, fixation of deleterious alleles linked to the demography of the artificial breeding, the purging of deleterious alleles in the same breeding context). Instead, we stress the fact that we focused on these genes due to the differentiation between wild and bred populations, which may serve as indicators of genetic erosion resulting from long-term mass rearing, and changes in the expression of these genes may be influence its behaviour or physical fitness. (Lines 446-448)

MINOR POINTS:

Line 33:

Please change “suppress” to “control”.

RESPONSE: We have corrected it accordingly. (Line 33)

Lines 34-35:

This sentence is only true in the case of augmentative biocontrol strategies. Please be more specific so as not to mislead people who are not specialists in biological control.

RESPONSE: We have corrected it accordingly. (Lines 34-38)

Line 37:

“developed or modified via artificial selection to meet human needs “: I don't think that's the usual way of doing things in the context of biocontrol. If I am wrong, please, provide appropriate references.

RESPONSE: We apologize for the inappropriate description. There are three primary biocontrol strategies in the field: classical biological control, augmentative biological control and conservation biological control. The method of using *A. gifuensis* as biocontrol agent belongs to augmentative biological control. Thus, we have revised this sentence to provide a more detailed explanation of augmentative biocontrol strategies, which introduces native or alien populations and reared indoors to meet the demands for immediate or longer-term control of a pest. (Lines 39-41)

Lines 40-42:

Again, you should better explain the context of your study, which focuses on

augmentative biological control. The problems outlined here do not apply to classical biological control where there is no wild population at the introduction site. I think it's important to clarify these points as the manuscript is not aimed solely at biological control specialists.

RESPONSE: Thanks for this suggestion. We have revised the text and added introduction about augmentative biological control. (Lines 39-41)

Lines 55-57:

Do you have any references for these statements?

RESPONSE: We added a reference to support our statements (Xie et al., 2020). (Line 79)

Lines 64-68:

Some of the “four questions” are not presented as questions. Please, try to be consistent.

RESPONSE: We changed all the four sentences to question style. (Lines 88-92)

Lines 74-78:

I'd really like to see a map showing the distribution of the three population types (N, B and T). This map could be included as additional data.

RESPONSE: We added a supplemental figure (Fig. S1) showing the geographic locations of sampled populations.

Lines 91-92:

The structure of this sentence suggests that SNPs are only used for evolutionary history inferences. This is (fortunately) not the case: please be clearer about the use of the different types of markers. Simply remove “respectively”.

RESPONSE: Thanks for the correction and we apologize for the misleading. We have revised these sentences as suggested. (Line 115)

Figure 1:

It is barely indicated in the figure legend that it corresponds to wild individuals only. Please change the title to "Population structure and demographic history of wild *A. gifuensis* populations..."

RESPONSE: We have corrected this legend accordingly.

Figure 1a:

The grey levels on the map seem unnecessary. I suppose it's the altitude, but why would this information be relevant? Please keep the map as simple as possible.

RESPONSE: We appreciate this valuable suggestion. We have removed the altitude information from the map.

Lines 95-96:

Please change “those from Clusters 1 and 2 were endemic to Yunnan Province “ to “those from Clusters 1 and 2 were found only in Yunnan Province”.

RESPONSE: We have corrected this sentence accordingly. (Line 118)

Lines 106-108:

This statement is not easy to visualise on the ADMIXTURE plot because you use province code in one case and population ID in the other. I suggest that you try to standardize the use of names of geographical locations or populations. At the very least, the population ID could be the same as the province code with an additional number to distinguish them.

RESPONSE: Thanks for this suggestion. We standardized the population ID. The first three letters were the same with the province code, while the fourth letter indicated different group of populations, including N (wild population), B (artificially bred population) and T (post-release population).

In addition, you mention "K=4" at the beginning of the sentence while you refer the reader not only to the ADMIXTURE figure but also to the PCA analysis figure (Fig. 1d), which is not related to a supervised analysis with a predetermined number of clusters. I suggest you remove the "K=4" information to let it only in the figure legend.

RESPONSE: We have removed the "K=4" information and only kept it in the figure legend. (Line 130)

Figure 1f:

Please, explain what the grey area is in the figure legend.

RESPONSE: The grey shadow denotes the last glacial maximum period,

approximately 11.5–20 K years ago.

Lines 118-120:

Lack of information to understand this statement. I guess the grey area is the LGM. If that is the case, your statement is not obvious to me. Please be more specific.

RESPONSE: We described that the grey area represents LGM.

Line 120:

“(LGM)”: I don't think you use this acronym anywhere else in the manuscript. Please remove.

RESPONSE: We have removed the abbreviation. (Line 145)

Lines 125-126:

“in each local population by each city”: Please be more specific and consistent. This description lacks clarity insofar as you use the terms "province code" or "population ID" elsewhere.

RESPONSE: Thanks for this question. We have revised this sentence. (Line 155)

In this section “*Environmental adaptation during the dispersal of wild populations*”, we calculated the AF by city.

In the latter section “*Natural selection restricts the post-release effects on wild populations*”, as we have described, AF were calculated at province level. This does not imply an inconsistency because in the latter section the YUN province was used to infer the ancestral state and only the non-YUN provinces (which do not have as sufficient sample size as YUN) were used to calculate the ΔAF . To avoid inaccurate ΔAF caused by insufficient alleles, we pooled all the individuals of each province in that section.

Figure 2a:

The label "Frequency" should not be use in the Y-axis. I guess this is a number or density of SNPs. Please change accordingly.

RESPONSE: Thanks for raising this point. We have changed “Frequency” to “Number of SNPs” in the plot according to the suggestion.

Line 139:

“nonsynonymous/synonymous (Nonsyn/Syn) ratios “: can you please describe this

statistic in more detail? I can't find a description of it in the methods section. I assume it is the ratio between the number of non-synonymous and synonymous polymorphisms within populations, often referred to as P_N/P_S . If this is the case, please state it clearly.

RESPONSE: Thanks for these comments. We apologize for the ambiguity, and we have added an explanation as follows: nonsynonymous/synonymous (Nonsyn/Syn) ratios (also known as P_N/P_S). (Line 177)

Supplementary Data 6:

Please, can you indicate the meaning of each column (with units) within the supplementary data?

RESPONSE: We added the meaning of each column at the end (bottom) of Supplementary Data 6.

Lines 158-160 and Figure 2f:

I find this statement lacking in justification. I would appreciate some statistical support for the idea, which is ultimately very visual, that the LG06 chromosome deserves to be looked at more closely.

RESPONSE: In the revised manuscript, we highlighted the 8 Mb region by red rectangle in Fig. 3b. We added statistical details to the Fig. 3b and its legend as follows: “The P values calculated between the 8 Mb region and the remaining genomic region were $P < 1e^{-200}$ for longitude, $P < 1e^{-200}$ for latitude, $P = 1.4e^{-9}$ for bio3, $P = 2.0e^{-149}$ for bio4, and $P < 1e^{-200}$ for bio5. P values were calculated from Wilcoxon rank sum tests”.

Lines 171-172:

"shared similar genetic structure" with what? Please, be more specific.

RESPONSE: We have revised this sentence as following: “ADMIXTURE and PCA results of artificially bred and post-release populations also revealed that the YUN populations were distinct from the peripheral populations (Fig. 1c and Fig. S5). The different structure pattern between YUN and the peripheral population was similar to wild population.” (Lines 228-232)

Line 182:

“(pairwise- $F_{ST} = 0.0121$)”: Is this a mean pairwise- F_{ST} . Please be more specific.

RESPONSE: The values is Weir and Cockerham's mean pairwise- F_{ST} between different population groups that was calculated using VCFtools. We have revised this

sentence accordingly. (Lines 240-243)

Line 197:

Please change “performed” to “performed”.

RESPONSE: We have revised this sentence. (Lines 255-256)

Lines 197-198:

Do you have a reference for this statement?

RESPONSE: We have added a reference (Xie et al., 2020). (Line 256)

Line 249:

"More profound" than what? What impacts? Please be more specific.

RESPONSE: Thanks for this question. The whole paragraph has been re-written. Please refer to our new description.

“Under random genetic drift (null hypothesis), strongly affected sites should show similar components (e.g., the Nonsyn/Syn ratio) with weakly affected sites (Fig. 5c). However, if natural selection tries to maintain the gene pool in wild population (ΔF_N), then we expect that the strongly affected sites should show lower fraction of Nonsyn sites compared to the weakly affected sites due to the fact that some Nonsyn mutations should be eliminated by purifying selection before they could reach a remarkable change in allele frequency (Fig. 5c). Under this selection pressure, the gene pool in wild population would be saved.” (Lines 339-345)

Figure 4e:

The nonsyn/syn values are lower for "weakly affected sites" than for "strongly affected sites". Can you explain this?

RESPONSE: Thanks for this question. If we understood correctly, the reviewer asked why the “simulated (expected)” Nonsyn/Syn ratios showed different distributions between “strongly affected sites” and “weakly affected sites” in the old Fig. 4e. We have extensively re-wrote that section and now the "weakly affected sites" than for "strongly affected sites" shared the same set of expected Nonsyn/Syn distribution estimated from random shuffling. The confusing point to the reader might be how the simulation (random shuffling) was performed. Together with a similar question below, we added the following explanations to the Methods section.

When evaluating the effect of the release on wild populations, we used the alteration in allele frequency (ΔAF) to quantify the impact on a particular population. For A.

gifuensis wild populations (N) in a particular non-YUN province, the present AF was denoted as AF_N , and the “change in AF” = $\Delta AF_N = AF_N - AF_A$, where AF_A was the AF in ancestral population inferred from YUN population plus outgroup species *A. ervi*. For example, $AF_A = 0$ means that all the alleles in YUN wild individuals and the *A. ervi* sequence were identical to the reference genome of *A. gifuensis*. For HEB and HEN provinces with presumably sufficient interaction between wild and released population, we selected the SNPs with $AF_A = 0 < AF_N$ and let $\Delta AF_N = AF_N - AF_A$. SNPs with $\Delta AF_N \geq 0.05$ were defined as affected sites. Among these affected sites, SNPs with significantly different AF between AF_N and AF_A were strongly affected sites and the remaining insignificant SNPs were weakly affected sites. Significance was defined by Fisher’s exact test on reference and alternative allele counts. *P* value was then adjusted by multiple testing correction to obtain an FDR. SNPs with $FDR < 0.05$ was defined as “strongly affected sites”.

During the simulation of the expected Nonsyn/Syn ratios, the AF_A and AF_N were randomized for 1,000 times for each province (HEB or HEN), respectively. Subsequently, $\Delta AF_N = AF_N - AF_A$ was calculated and the “strongly affected sites” and “weakly affected sites” were defined. The expected Nonsyn/Syn ratio was calculated accordingly. Each round of randomization in each province will produce one Nonsyn/Syn ratio. For each province of HEB, HEN, or pooled non-YUN, the randomization was done for 1,000 times.

Line 308:

“the genetic drift of inbreeding”: this is a peculiar expression. I guess "genetic drift" would be sufficient here.

RESPONSE: We have corrected it as suggested. (Line 426)

Lines 308-309:

“individuals within populations were genetically more similar to one another than individuals from external populations”: A very circular phrase, in that the notion of genetic proximity between individuals is an integral part of the very definition of what a 'population' is. The whole sentence is of little interest, as it merely states the obvious.

RESPONSE: Thanks for this comment. We have revised this sentence. The primary aim of this section was to discuss the impact of extensive inbreeding, resulting in distinct genetic compositions between artificially bred populations and wild populations. (Lines 426-429)

Line 309:

Please change “t” to “to”.

RESPONSE: Corrected accordingly. (Lines 428-429)

Lines 313-315:

Unclear. Please rephrase.

RESPONSE: We have corrected it accordingly. “Additionally, a higher level of LD has been found during the domestication process of economically important animals and plants. For the parasitoid wasp *A. gifuensis*, despite the frequent introduction of wild populations to expand the artificially bred populations, the latter still exhibited stronger LD compared to the former.” (Lines 432-435)

Lines 319-321:

Again, I would like a reference for this statement.

RESPONSE: We have added a reference (Xie et al., 2020). (Line 443)

Line 340:

“selected genes”: This is not clear. You refer to "genes under selection", is that it? Anyway, what you describe as genetic erosion is mainly driven by genetic drift, not selection, so you should be more careful with the terms you use.

RESPONSE: Thanks for this reminder. As described above, we have modified the description of the “selected genes” to “differentiated genes identified under the comparison between artificially bred and wild populations”, and this term will be compatible to both genetic drift or natural selection. (Lines 446-448)

Lines 342-344:

This may seem a reasonable proposition, but how does it relate to the results presented in this manuscript? Firstly, the impact of biological control on the wild gene pool is considered to be of little importance here. Second, none of the results presented here indicate whether the genetic composition of breed individuals is detrimental in the very peculiar context of breeding environments and human agricultural activity (i.e. rapid and effective control of pest populations does not necessarily require the same traits as in the wild).

You refer to the risks for survival performance, but is there any signs of any problems with the breeding of this species?

RESPONSE: Indeed, the environment in artificially bred populations differs from that in the wild, and we plan to conduct further research to explore the genetic variations that have adapted to captive environment. It has been found that long-term mass rearing

has resulted in rapid population degradation, decreased fecundity, and reduced parasitoid performance in artificially bred *A. gifuensis* populations (Xie et al., 2020) (which is, what this reviewer called “detrimental”). And our results also demonstrated that artificially bred populations performed lower genetic diversity than wild populations. In this section, we discussed the genetic rescue method for recovery of genetic diversity of mass rearing populations.

Line 355:

“severely affected”: I'm not a big fan of this term, which seems to me to refer mainly to a decline in diversity. Please be more precise in the terms you use.

RESPONSE: Thanks for this comment. We changed “severely” to “strongly”. (Lines 510 and 514)

Lines 388-390:

On what basis do you consider it to be a good model? It's certainly an interesting case study, but why "model"? I suppose you could consider the species as a model if you had a large ancestral sample, significant genomic resources, and the species was particularly easy to sample. As all these conditions are not met, I would avoid using this term, which unnecessarily overstates the results of the study.

RESPONSE: To avoid overstatement, we used “system” instead of “model”. (Line 553)

Lines 393-404:

Thank you for clarifying here that these are only females (see line 74). This is important because it has a strong impact on sample size, as it seems to me that the species is haplodiploid.

RESPONSE: Thanks for this recognition. We took into consideration the haplodiploid sex determination system of *A. gifuensis* and chose diploid females for its population genetic studies. We also clarified this reason in the Methods section. (Lines 558-559)

Line 396:

What are the criteria for defining "areas without release records"? In particular, is there a minimum distance between a known release site and such areas for them to be considered as such? If such a criterion exists, is it based on the known dispersal characteristics/abilities of the species?

RESPONSE: We added the information about sampling collection and the definition

of wild populations in the Methods section. Wild individuals were collected far away from areas where *A. gifuensis* was used for biocontrol, considering the flight distance of mummified alates of *M. persicae* (~2.16 km) as an indicator of parasitoids dispersal. The distance between wild and post-release populations was from 186.27 km to 11,807.19 km, which calculated based on longitude and latitude. (Lines 561-566)

Lines 402-403:

Why do you say "additional samples"? As I understand it, these 162 individuals are part of the populations B, T and N presented above. The whole sentence can be deleted to avoid confusion. Or it could simply be pointed out that Yunnan is over-represented in the sample for historical reasons. Incidentally, these historical reasons are not clear to me: is the species more abundant in this province?

RESPONSE: Thank you for the comments. We deleted this sentence to avoid confusion.

Line 450:

Please remove "Thus".

RESPONSE: Corrected it accordingly. (Line 624)

Line 457:

Please write "4.FST"

RESPONSE: Corrected it accordingly. (Line 632)

Lines 462-463:

How were the K shown in Figure 1c selected? If the choice was made by the authors on the basis of visual criteria, it is important to present the figures with all values of K as additional information.

RESPONSE: Thanks for your suggestion. We added a supplementary figure (Fig. S2) about ADMIXTURE cross-validation error of all K values and population structure with other K value.

Line 477:

"MCMC": don't you mean MSMC?

RESPONSE: We apologize for the Typo. We have corrected it accordingly. (Line 656)

Line 478:

Please change “in the bumblebee” to “as estimated for the bumblebee”.

RESPONSE: Corrected it accordingly. (Line 657)

Line 479:

Do you have any reference for this? 15 generations seems like a lot.

RESPONSE: We have added these references (Xin, 1986; Zhao et al., 1980). (Line 658)

Lines 511-512:

“we selected 35 wild populations (N)”: As I understand it, the whole wild sample is made up of 35 populations (supplementary data 1), isn't that right? Shouldn't it be "we used the 35 wild populations (N)"?

RESPONSE: Yes, the 35 wild populations were the whole wild samples, and we corrected the sentence accordingly. (Line 712)

Lines 523-524:

“ $AF_A < 0.01$ ”: According to Supplementary Data 1, the maximum number of individuals is 14, which gives a minimum possible frequency of $1/28$, in other words 0.036. This threshold of 0.01 corresponds to populations with an estimated allelic frequency of 0, doesn't it? If so, this criteria is somewhat misleading. Why not simply say that you are considering SNPs whose frequencies are equal to 0 in the YUN populations? However, it is possible that the AFs are calculated on all pooled samples. If this is the case, you need to be much clearer, particularly in the use of terms, for example by a better use of the terms “province” and “population”, as in Supplementary Data 1 and 2.

RESPONSE: We apologize for the obscure expression. In the previous manuscript, the totally 67 individuals in YUN province were used to infer the ancestral AF_A . The minimum AF_A value was $1/134 = 0.0075$. In the revised manuscript, we have used all YUN population plus outgroup *A. ervi* to infer the ancestral state so that we used the stringent criteria and no longer allowed derived alleles at these sites in YUN province. For example, $AF_A = 0$ means that all the alleles in YUN wild individuals and the *A. ervi* sequence were identical to the reference genome of *A. gifuensis*.

Lines 526-527:

Again, so the AF is measured at the Province level? Or at the population level? This is unclear in the whole AF description section. Please be more specific.

RESPONSE: We apologize for the ambiguous definition. As we have answered in the previous questions, the AF is measured at the province level. The corresponding section in the manuscript has been extensively re-written.

Line 528:

Please change “number of sites” to “number of SNP sites”.

RESPONSE: These sentences have been revised.

Line 538:

I'm not sure I understand what you mean by "simulation". Can you elaborate?

RESPONSE: Thanks for this question. We re-phrased the corresponding Methods. As we have responded to the previous question, we added the following description to the Methods section.

When evaluating the effect of the release on wild populations, we used the alteration in allele frequency (ΔAF) to quantify the impact on a particular population. For *A. gifuensis* wild populations (N) in a particular non-YUN province, the present AF was denoted as AF_N , and the “change in AF” = $\Delta AF_N = AF_N - AF_A$, where AF_A was the AF in ancestral population inferred from YUN population plus outgroup species *A. erwi*. For example, $AF_A = 0$ means that all the alleles in YUN wild individuals and the *A. erwi* sequence were identical to the reference genome of *A. gifuensis*. For HEB and HEN provinces with presumably sufficient interaction between wild and released population, we selected the SNPs with $AF_A = 0 < AF_N$ and let $\Delta AF_N = AF_N - AF_A$. SNPs with $\Delta AF_N \geq 0.05$ were defined as affected sites. Among these affected sites, SNPs with significantly different AF between AF_N and AF_A were strongly affected sites and the remaining insignificant SNPs were weakly affected sites. Significance was defined by Fisher's exact test on reference and alternative allele counts. *P* value was then adjusted by multiple testing correction to obtain an FDR. SNPs with $FDR < 0.05$ was defined as “strongly affected sites”. (Lines 720-735)

During the simulation of the expected Nonsyn/Syn ratios, the AF_A and AF_N were randomized for 1,000 times for each province (HEB or HEN), respectively. Subsequently, $\Delta AF_N = AF_N - AF_A$ was calculated and the “strongly affected sites” and “weakly affected sites” were defined. The expected Nonsyn/Syn ratio was calculated accordingly. Each round of randomization in each province will produce one Nonsyn/Syn ratio. For each province of HEB, HEN, or pooled non-YUN, the randomization was done for 1,000 times. (Lines 749-755)

Line 544:

“in comparison to *Aphidius ervi*”: I assume you're referring to the *Aphidius ervi* genome. Please be more specific and indicate which version of the genome you are using.

RESPONSE: We have added the detail referring information of the *Aphidius ervi* genome (Bioinformatic Platform for Agrosystem Arthropods: https://bipaa.genouest.org/sp/aphidius_ervi/download/genome/v3.0/). (Lines 758-760)

Again, we thank you for your time and effort in reviewing this manuscript.

Reviewers' Comments:

Reviewer #1:

Remarks to the Author:

In this manuscript, Bingyan Li et al. report a population genomic survey of the aphid parasitoid *Aphidius gifuensis* in China to determine whether the release of mass-reared parasitoid wasps may alter the genetic features of their natural populations. The authors are particularly interested in studying the effects of inbreeding on parasitoid performance at the field scale, and they looked for genomic evidence to support that idea. The survey included samples from wild and cage-reared populations and samples from a *Myzus persicae* population currently managed using biological control. The authors found a genetic signature of environmental adaptation during the dispersal of wild populations from the zone with the higher genetic diversity to other zones in China, with no significant loss of genetic diversity. The authors also found that caged-reared parasitoid populations show decreased genetic diversity. However, no genetic effects on wild populations were observed after their release as biological control agents.

In this new version, the authors have addressed all the reviewers' comments, and I feel they have answered all of them reasonably. Significant improvements were made in the manuscript accordingly. They have toned down data interpretation and discussion, and putative selection agents acting on populations are now explicitly discussed; indeed, the role of the interactions with other organisms and the environment is now discussed. Consequently, I feel this manuscript is now acceptable in Nature Communications after addressing the following minor points.

Lines according to the annotated in red version:

Line 255: Please explain what you mean by capacity and fecundity. Some authors refer to infectivity (preference) and virulence (a proxy of parasitoid fitness) when they measure parasitoid performance, as in Zepeda-Paulo et al. (2013).

Line 266: Do you have some parasitoid performance data from the studied populations? It could be interesting to test whether genetic differences between populations correlate with the performance of the same aphid host (i.e., *Myzus persicae*).

Line 284: To avoid confusion, I suggest being precise regarding parasitoid wasps collected from host aphids feeding on tobacco or other plants. I assume that all wasps were collected from *Myzus persicae* mummies, but please indicate if other aphids were also collected.

Line 286: Fluctuations and challenges may include using other host aphid species, as *A. gifuensis* is known to be a generalist parasitoid wasp. See:

Pan, M. Z., & Liu, T. X. (2014). Suitability of three aphid species for *Aphidius gifuensis* (Hymenoptera: Braconidae): Parasitoid performance varies with hosts of origin. *Biological Control*, 69, 90-96.

Line 290: Replace "host plants" with "aphid host plant".

Line 341: Delete "tries to"

Lines 398-400: I suggest being cautious, as your results are restricted to *A. gifuensis* populations only collected from *Myzus persicae*. The impact of releasing this parasitoid wasp must be tested on populations using other aphid hosts. Also, this generalization still needs empirical evidence from bioassays that test parasitoid performance (e.g., infectivity, virulence, host fidelity) is not significantly affected by introgressions into wild populations and populations using other aphid hosts.

Line 442: Replace "aphid" with "M. persicae populations".

Line 473: Replace "host plant" with "aphid host plant".

Line 521: Replace "fed on" with "parasitizing M. persicae aphids on".

Line 522: Include "from aphid mummies on" after "individuals were collected".

Line 523: Include "from aphid mummies on" after "individuals were collected".

Reviewer #2:

Remarks to the Author:

I read the authors' replies to my reviews in detail. I have to say that I found most of the answers convincing and most of my questions answered. The authors have thoroughly revised the manuscript to make it clearer, and they are also more cautious in the way they interpret the results. The addition of some new figures is welcome, such as Figures 5a, 5b and 5c, which are very helpful in understanding the authors' approach.

I am therefore very happy with the new version of the manuscript. I have only two minor points to make.

The first concerns the allele polarisation. In the manuscript (lines 635-368 of the final version) the authors seem to have used *A. ervi* to polarise the allele. However, in the pdf response to the reviewer, there is some mention of using both the YUN population and *A. ervi* at the same time. For example, when the authors say "[...] where AFA was the AF in the ancestral population inferred from the YUN population plus outgroup species *A. ervi*". What does this mean in the end? Can the authors assure me that only *A. ervi* was used to polarise the alleles? And if so, was it possible to polarise all the SNPs? I would like some more information about that.

The second point concerns the use of the term "simulation" in line 655, which still doesn't seem clear to me and doesn't seem to correspond to the most common use of the term. This word is only used once in the whole manuscript, which suggests that it could easily be removed.

RESPONSE TO REVIEWER COMMENTS

In this manuscript, Bingyan Li et al. report a population genomic survey of the aphid parasitoid *Aphidius gifuensis* in China to determine whether the release of mass-reared parasitoid wasps may alter the genetic features of their natural populations. The authors are particularly interested in studying the effects of inbreeding on parasitoid performance at the field scale, and they looked for genomic evidence to support that idea. The survey included samples from wild and cage-reared populations and samples from a *Myzus persicae* population currently managed using biological control. The authors found a genetic signature of environmental adaptation during the dispersal of wild populations from the zone with the higher genetic diversity to other zones in China, with no significant loss of genetic diversity. The authors also found that caged-reared parasitoid populations show decreased genetic diversity. However, no genetic effects on wild populations were observed after their release as biological control agents.

In this new version, the authors have addressed all the reviewers' comments, and I feel they have answered all of them reasonably. Significant improvements were made in the manuscript accordingly. They have toned down data interpretation and discussion, and putative selection agents acting on populations are now explicitly discussed; indeed, the role of the interactions with other organisms and the environment is now discussed. Consequently, I feel this manuscript is now acceptable in Nature Communications after addressing the following minor points.

RESPONSE: Thank you very much for detail reviewing and comments about our manuscript. We further revised the main text according to your suggestions and described our responses in this letter.

Lines according to the annotated in red version:

Line 255: Please explain what you mean by capacity and fecundity. Some authors refer to infectivity (preference) and virulence (a proxy of parasitoid fitness) when they measure parasitoid performance, as in Zepeda-Paulo et al. (2013).

RESPONSE: We described the effect of generations of mass rearing on phenomena of *A. gifuensis* in detail, such as reducing reproduction, decreased emergence rate and decreased adult longevity. (Lines 239-240)

Line 266: Do you have some parasitoid performance data from the studied populations? It could be interesting to test whether genetic differences between populations correlate with the performance of the same aphid host (i.e., *Myzus persicae*).

RESPONSE: Thanks for raising this good suggestion. Indeed, we lacked the behavioral measurements and statistics about these parasitoid performance data of the

studied populations, which currently prevents us from testing the correlation between genotype and phenotype. However, the results and conclusions of this study will guide our future studies on exploring the genetic differences between populations correlated with parasitoid performances.

Line 284: To avoid confusion, I suggest being precise regarding parasitoid wasps collected from host aphids feeding on tobacco or other plants. I assume that all wasps were collected from *Myzus persicae* mummies, but please indicate if other aphids were also collected.

RESPONSE: We revised this sentence by specifying the host aphid (*Myzus persicae*) from which these populations were collected to enhance accuracy. (Lines 266-268)

Line 286: Fluctuations and challenges may include using other host aphid species, as *A. gifuensis* is known to be a generalist parasitoid wasp. See:
Pan, M. Z., & Liu, T. X. (2014). Suitability of three aphid species for *Aphidius gifuensis* (Hymenoptera: Braconidae): Parasitoid performance varies with hosts of origin. *Biological Control*, 69, 90-96.

RESPONSE: Thanks for this suggestion. We added the fluctuations and challenges of using other host aphid species on *A. gifuensis*. (Line 270)

Line 290: Replace “host plants” with “aphid host plant”.

RESPONSE: We have corrected it accordingly. (Line 273)

Line 341: Delete "tries to"

RESPONSE: We have corrected it accordingly. (Line 314)

Lines 398-400: I suggest being cautious, as your results are restricted to *A. gifuensis* populations only collected from *Myzus persicae*. The impact of releasing this parasitoid wasp must be tested on populations using other aphid hosts. Also, this generalization still needs empirical evidence from bioassays that test parasitoid performance (e.g., infectivity, virulence, host fidelity) is not significantly affected by introgressions into wild populations and populations using other aphid hosts.

RESPONSE: We revised this sentence with the background about the mass rearing and biocontrol application history of *A. gifuensis* on *M. persicae* in China, and made the conclusion more cautiously. We acknowledged the challenges, limitations, and future efforts you mentioned. (Lines 346-353)

Line 442: Replace “aphid” with “*M. persicae* populations”.

RESPONSE: Corrected it accordingly. (Line 388)

Line 473: Replace “host plant” with “aphid host plant”.

RESPONSE: Corrected it accordingly. (Line 419)

Line 521: Replace "fed on" with “parasitizing *M. persicae* aphids on”.

RESPONSE: Corrected it accordingly. (Line 463)

Line 522: Include “from aphid mummies on” after “individuals were collected”.

RESPONSE: Corrected it accordingly. (Line 464)

Line 523: Include “from aphid mummies on” after “individuals were collected”.

RESPONSE: Corrected it accordingly. (Line 465)

RESPONSE TO REVIEWER COMMENTS

I read the authors' replies to my reviews in detail. I have to say that I found most of the answers convincing and most of my questions answered. The authors have thoroughly revised the manuscript to make it clearer, and they are also more cautious in the way they interpret the results. The addition of some new figures is welcome, such as Figures 5a, 5b and 5c, which are very helpful in understanding the authors' approach.

I am therefore very happy with the new version of the manuscript. I have only two minor points to make.

Response: Thank you very much for your acknowledgement.

The first concerns the allele polarisation. In the manuscript (lines 635-368 of the final version) the authors seem to have used *A. ervi* to polarise the allele. However, in the pdf response to the reviewer, there is some mention of using both the YUN population and *A. ervi* at the same time. For example, when the authors say "[...] where AF_A was the AF in the ancestral population inferred from the YUN population plus outgroup species *A. ervi*". What does this mean in the end? Can the authors assure me that only *A. ervi* was used to polarise the alleles? And if so, was it possible to polarise all the SNPs? I would like some more information about that.

Response: Thanks for raising this question. We apologize for the confusion. We would like to clarify the scheme with a new diagram (Fig. S12):

In the correlation tests between SNPs and regional factors, all SNPs in all regions (including YUN) were used. For a specific bi-allelic SNP site in the *A. gifuensis* population, either the reference allele or the alternative allele is identical to the *A. ervi* sequence (Fig. S12a). In the former case, the reference allele of *A. gifuensis* should be the ancestral allele, then derived allele frequency (DAF) = alternative allele frequency AF. In the latter case, the alternative allele in *A. gifuensis* is likely to be the ancestral allele and thus $DAF = 1 - AF$ (Fig. S12a). In the previous version of manuscript, we explained that in the latter case, AF is changed to $1 - AF$. (Lines 648-653)

In the later part involving ΔAF , only the non-YUN provinces were investigated. The AF in a specific non-YUN wild population (denoted as AF_N) is exact the same as we previously defined (Fig. S12a). However, to ensure that the AF_N truly represents ΔAF (the change in AF), we selected the sites with the ancestral $AF_A = 0$. It means that all the alleles in YUN wild individuals and the *A. ervi* sequence were identical, thus representing the ancestral allele (Fig. S12b). In other words, for this analysis, not all SNPs in a non-YUN province were used; rather, the AFs of the specific SNPs of interest were exactly identical to those defined in the correlation test (Fig. S12a). The rationale for exclusively focusing on SNPs with ancestral $AF_A = 0$ was that in other cases, it becomes challenging to ascertain the AF_A , making it difficult to determine the change in AF_N . However, in the correlation tests between SNPs and regional factors, we do not interrogate the change in AF within a specific time scale, eliminating computational challenge in this regard. (Lines 662-672)

The second point concerns the use of the term "simulation" in line 655, which still doesn't seem clear to me and doesn't seem to correspond to the most common use of the term. This word is only used once in the whole manuscript, which suggests that it could easily be removed.

Response: Thanks for the reminder. We changed “simulation” to “calculation”. (Line 680)

Reviewers' Comments:

Reviewer #1:

Remarks to the Author:

I have no further comments. Congratulations to the authors for their work.

Reviewer #2:

Remarks to the Author:

The previous version of the manuscript was already very satisfactory from my point of view. The current version seems to me to be a quality article in the field. The authors' response regarding allele polarisation provides a pedagogical clarification of a point that is often overlooked in other similar studies.

I have also carefully read the responses to the second reviewer's comments and find them equally satisfactory.